# *acorde* unravels functionally interpretable networks of isoform co-usage from single cell data

Angeles Arzalluz-Luque [1,2], Pedro Salguero [1], Sonia Tarazona[1,4 ✉] & Ana Conesa [2,3,4 ✉]

Alternative splicing (AS) is a highly-regulated post-transcriptional mechanism known to modulate isoform expression within genes and contribute to cell-type identity. However, the extent to which alternative isoforms establish co-expression networks that may be relevant in cellular function has not been explored yet. Here, we present *acorde*, a pipeline that successfully leverages bulk long reads and single-cell data to confidently detect alternative isoform co-expression relationships. To achieve this, we develop and validate percentile correlations, an innovative approach that overcomes data sparsity and yields accurate co-expression estimates from single-cell data. Next, *acorde* uses correlations to cluster co-expressed isoforms into a network, unraveling cell type-specific alternative isoform usage patterns. By selecting same-gene isoforms between these clusters, we subsequently detect and characterize genes with co-differential isoform usage (coDIU) across cell types. Finally, we predict functional elements from long read-defined isoforms and provide insight into biological processes, motifs, and domains potentially controlled by the coordination of post-transcriptional regulation. The code for *acorde* is available at https://github.com/ConesaLab/acorde.

[1] Department of Applied Statistics, Operations Research and Quality, Universitat Politècnica de València, Valencia, Spain. [2] Institute for Integrative Systems Biology (CSIC-UV), Spanish National Research Council, Paterna, Valencia, Spain. [3] Microbiology and Cell Sciences Department, Institute for Food and Agricultural Research, University of Florida, Gainesville, FL, USA. [4] These authors contributed equally: Sonia Tarazona, Ana Conesa. ✉email: sotacam@eio.upv; ana.conesa@csic.es

Single-cell RNA-seq (scRNA-seq) has revolutionized transcriptomics analysis, especially as the development of technologies with increasingly higher throughputs has enabled the processing of thousands of single cells simultaneously, boosting the amount of biological diversity that can be captured in a sequencing experiment[1]. The technology has been extensively applied to the discovery of new cell types and the characterization of their transcriptional profiles, resulting in the definition of cell type marker genes[2–7]. Even though scRNA-seq datasets can encode several levels of granularity in the form of cell subtypes, these studies rely on the low number of features required to recapitulate the cell type structure of the data[8], which situates cell type characterization efforts at the baseline of understanding the intricacy of single-cell biology. scRNA-seq studies have also tackled transcriptional dynamics and how they relate to cell type properties. These include methods for the study of dynamic processes, namely pseudotime[9,10] and RNA-velocity[11] analyses, which have provided insight on cell differentiation and the mechanisms behind cell state transitions[7,12–16]. Moreover, novel methods have been developed to allow the inference of gene regulatory networks (GRNs) at the single-cell level[17], in an attempt to combine single-cell information with extant knowledge and infer relationships between genes and transcriptional regulators at a higher resolution.

Single-cell research is nevertheless far from realizing its full potential. On the contrary, the timing is now optimal for the field to undertake the investigation of deeper layers of cellular complexity. In particular, the investigation of alternative splicing (AS) and isoform expression dynamics has remained a challenge to the field. Reasons for this include the uncertainty of short read-based isoform quantification, which is exacerbated by the lower number of available reads per transcript in comparison to bulk libraries[18], and the fact that the most popular scRNA-seq methods are heavily 3′ end biased, which precludes the unambiguous identification of alternative transcript variants[19]. Current methods for the study of isoforms in single cells, therefore, rely on alternative metrics that either avoid isoform-level expression estimation completely[20,21] or exclusively consider individual splicing events[22–26], generally leaving isoform characterization aside, with some recent exceptions[27]. Meanwhile, long read RNA sequencing (lrRNA-seq) of single cells is beginning to emerge an alternative approach to mitigate this ambiguity, given that it successfully grasps how individual events are combined into alternative isoforms[19]. Long read studies have expanded the field's notions of cell type-specific splicing from event inclusion towards isoform selection patterns and showed that cell type-specific isoform expression can be detected in both broad types as well as cell subtypes[28–32]. Unfortunately, the sequencing depth constraints intrinsic to long read protocols[19] have limited the amount of isoform diversity that can be captured by single-cell long read transcriptomics[28–30], and datasets generally show low levels of redundancy between cells of the same cell type.

Notwithstanding this limited scenario, there are a number of relevant questions regarding the importance of splicing for cell identity and function that can only be resolved by evaluating isoform expression at the single-cell level. In fact, splicing differences have been shown to discriminate cell types with an accuracy comparable to that obtained using gene expression[33] while integrating AS and gene expression changes has led to the discovery of cell subtypes and states that were otherwise not detected[27,34–36]. Especially relevant among these inquiries is the much-debated issue of whether individual cells express one or several isoforms, that is, whether the isoform diversity observed in bulk studies is recapitulated by each single cell or, alternatively, arises as a result of multiple cells expressing one of the gene's isoforms. Ever since the publication of the first scRNA-seq studies, short reads have been used to answer this question, usually via the characterization of splicing event—rather than isoform expression- modalities. Successive studies have provided non-conclusive results, with evidence of bimodal splicing patterns[22,37,38] as well as concerns regarding the relationship between bimodal isoform detection and technical noise[18,39], a controversy that suggests that new approaches are needed to understand the isoform landscape of single cells.

Another pending question for the field is whether isoform expression programs involve co-expression relationships between transcript variants from different genes. So far, the application of long read technologies to single-cell data has served to unravel coordinated event choice patterns within isoforms of the same gene[40,41], however, cross-gene isoform expression networks have not been investigated. In other words, there have been no studies addressing potential codependency between genes regarding the selection of transcript variants from their isoform repertoire, or the implications of this coordination for cell-state and cell-type properties. This is not only related to the general constraints of single-cell isoform studies, but also to the lack of computational methods and mathematical models to extract this complex signal from the data. In spite of the present research gap, isoform co-expression networks are an anticipated consequence of the regulation of splicing by RNA binding proteins (RBPs) and other splicing factors, and their investigation constitutes an opportunity to gain insight on the functional role of AS. Moreover, a multiple cell type and high cell throughput context such as that of scRNA-seq data constitutes a far more suitable data scenario to that of bulk RNA-seq. Given this context, this is undoubtedly a timely inquiry to make.

In the present study, we hypothesize that isoform expression coordination exists as a result of AS regulation, and that it can be computationally detected in the form of isoform groups showing co-variation across cell types. To demonstrate this hypothesis, we have designed an end-to-end, data-intensive pipeline for the study of isoform networks (Fig. 1). First and foremost, we employed a hybrid strategy where bulk long-reads and single-cell Illumina sequencing were integrated to estimate isoform expression at the single cell level. To unlock the limitations of extant correlation metrics in the single-cell context[42], we developed a strategy to obtain noise-robust correlation estimates in scRNA-seq data, and a semi-automated clustering approach to detect modules of co-expressed isoforms across cell types. We additionally defined and implemented Differential Isoform Usage (DIU) and co-Differential Isoform Usage (coDIU) analyses in order to leverage the multiple cell types contained in single-cell datasets. Finally, to couple these analyses with a biologically interpretable readout, we incorporated a functional annotation step in which several databases and prediction tools were integrated to add isoform-specific functional information (Fig. 1, Supplementary Note).

We have hereby applied this pipeline (Fig. 1) to the analysis of two publicly available mouse neural datasets, including scRNA-seq Smart-Seq2 (primary visual cortex, generated by Tasic et al.[43,44]) and bulk ENCODE PacBio long-read data (mouse cortex and hippocampus, generated by Wyman et al.[45]). As a result, we successfully detected cell type-level co-expression of isoforms in a manner that was independent of gene-level expression. Furthermore, we demonstrated that these isoforms encode shared functional properties, highlighting the role of post-transcriptional processing as in the fine-tuning of cellular functions and the encoding of cell type identity. This pipeline has been implemented in the R package *acorde* (https://github.com/ConesaLab/acorde).

## Results

**Enabling multi-group differential expression (DE) of isoforms in single-cell data.** Traditionally, RNA-seq studies have used

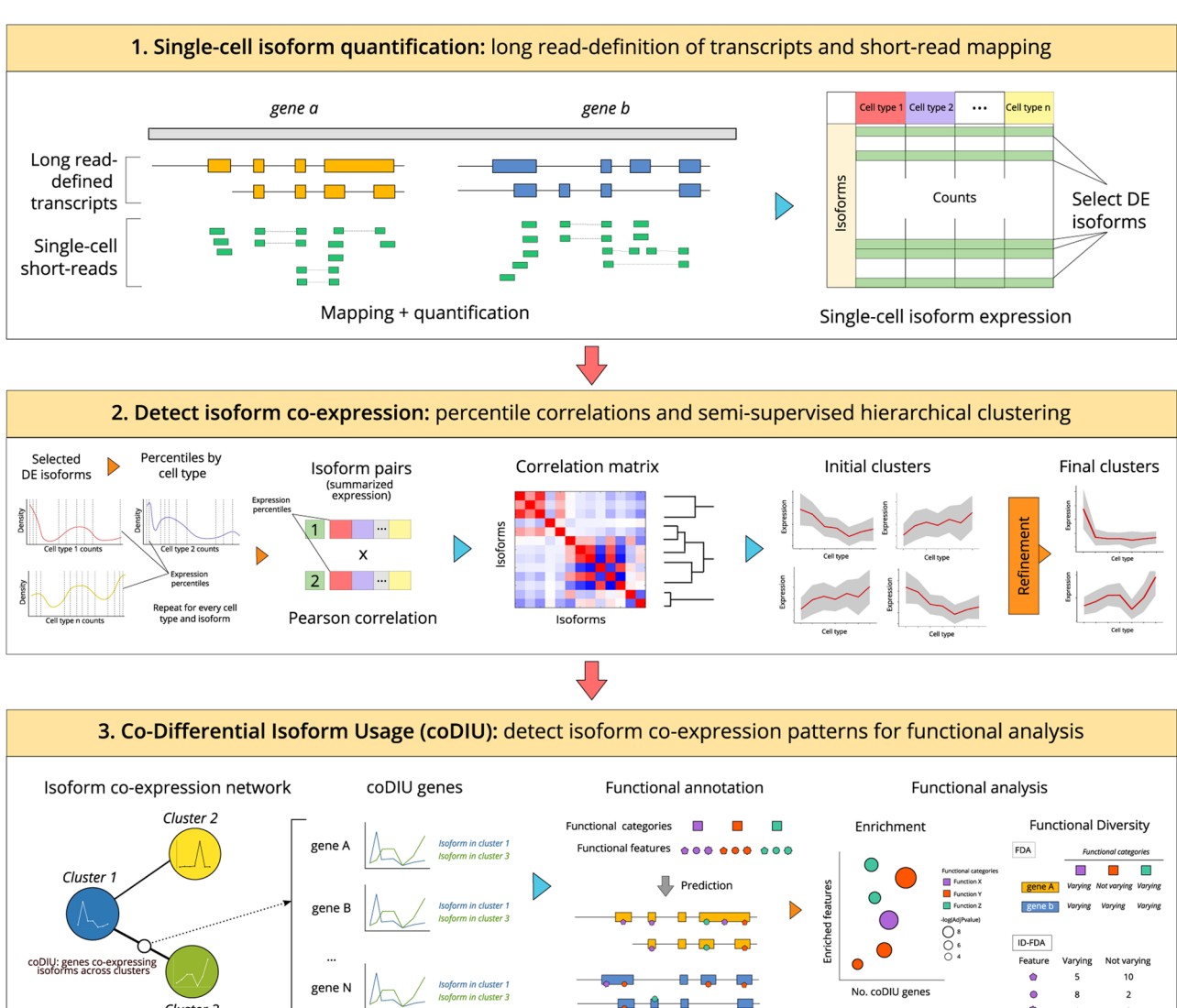

**Fig. 1 *acorde* workflow.** The *acorde* pipeline includes three main analysis modules. First, long read RNA-Seq data is used to define isoform models and short, single-cell RNA-Seq reads are mapped to the long read-generated transcriptome. Isoform are then tested for multi-group differential expression and those that are significantly DE in at least one of the cell types are selected. Next, percentile correlations are computed to cluster isoforms with similar expression patterns across cell types. Finally, gene pairs are tested for co-differential isoform usage, detecting genes that form co-expression relationships for subsequent functional analysis.

publicly available reference transcriptomes such as RefSeq and ENSEMBL for short read isoform quantification. However, most tissues and cell types will express only a subset of the genes and isoforms contained in the reference, with previous studies showing that isoform detection accuracy increases when adopting tissue-specific isoform sets as a reference for mapping[46]. Long read technologies have the potential to achieve this, while also expanding the reference with novel isoforms that have not yet been annotated[47]. Given the low depth of single-cell long read datasets[19], we employed bulk PacBio long-reads obtained from ENCODE[45] to define a mouse, neural-specific transcriptome, that, after extensive curation using the SQANTI3 toolkit (https://github.com/ConesaLab/SQANTI3) contained 36,986 isoforms belonging to 12,692 genes (see Supplementary Note).

To quantify the expression of the long read-defined isoforms at the single-cell level, we made use of a publicly available, deeply sequenced, full-length, short-read single-cell RNA-seq dataset by Tasic et al.[43]. In total, 1591 cells and 16,240 isoforms from 8814 genes were retained after quality control (see "Methods"). Using

the labels from the original characterization of the dataset, we assigned cells to 7 broad cell types, 5 glial (microglia, endothelial cells, oligodendrocytes, oligodendrocyte precursor cells (OPCs) and astrocytes) and 2 neural (GABA-ergic and glutamatergic neurons), each of which can be divided into several, distinct subtypes (Supplementary Fig. 1a). Next, to select isoforms with robust co-variation and non-constitutive expression, a multi-group strategy was used to detect isoforms showing DE in at least one cell type, combining the ZinBWaVE zero-expression weighting strategy[48] with bulk-designed DE methods DESeq2[49] and edgeR[50] (see "Methods").

Of note, the Tasic dataset presented a drastic cell number imbalance between neural (~720 cells/cell type) and glial cell types (~30 cells/cell type), which resulted in the underestimation of transcriptional differences between non-neural types (Supplementary Fig. 1b). To balance sample sizes, we performed 50 rounds of random neural cell sampling ($n = 45$ cells) followed by DE testing using both edgeR and DESeq2 (see "Methods"). Although DESeq2 proved to be slightly more robust across

independent runs than edgeR, Jaccard index values indicated that the majority of isoforms were consistently detected as DE (DESeq2: mean no. of DE isoforms = 6908 ± 101, mean Jaccard index = 0.84 ± 0.02; edgeR: mean no. of DE isoforms = 6016 ± 410, mean Jaccard index = 0.74 ± 0.02). In addition, this strategy revealed that considering the union of edgeR and DESeq2 results contributed to improve robustness (mean no. of DE isoforms in union between methods = 9399 ± 248, Jaccard index = 0.82 ± 0.01). Upon testing, we selected a consensus set of isoforms consisting in those transcripts detected as significant (FDR < 0.05) by at least one DE method in ≥50% downsampling runs. This consensus set included 9393 isoforms from 4223 genes, however, minor isoforms (those accumulating <10% of total gene expression) were additionally filtered, resulting in the removal of ~10% of the consensus DE set (969 isoforms). Finally, since two or more isoforms with differential cell type expression are required to form co-splicing relationships (see "Methods"), transcripts with no other same-gene DE counterpart were removed, retaining 6794 isoforms from 2696 genes for downstream analysis.

**Detecting isoforms showing cell type-level co-expression.** Co-expression signals in single-cell data are weak and have often resulted in a poor performance of traditional correlation and network inference methods[42,51]. Although data transformation approaches[52] and alternative metrics[42] have been proposed, these are more complex to apply and considerably less interpretable, respectively. Furthermore, most of these studies have only investigated gene-level co-expression[17], often ignoring the AS regulatory landscape. To address these limitations, we implemented a *percentile correlation* strategy: a simple, scalable approach to overcome single-cell noise in isoform co-expression studies (Fig. 2a).

Our approach considers cell-type identity to be defined by context-specific gene expression and within-cell type stochasticity to arise from a combination of technical noise[53,54] and biological mechanisms such as transcriptional bursting[55]. These translate into the sparse and heterogeneous expression patterns typically observed in scRNA-seq (Supplementary Fig. 2a) and result in high variance across all levels of expression (Supplementary Fig. 2b). Together, these effects mask the co-expression signal in the data and tend to yield low correlation values when using traditional metrics (Fig. 2b). To overcome this problem, we treat single cells of the same cell-type as biological replicates, that is, instances that represent the state of a delimited cell population, but are differently affected by the aforementioned combination of technical and biological forces. In this context, the expression distribution of any given isoform across the population can be considered to be the signature of the isoform in that cell-type. To translate these assumptions into a metric, the expression of each isoform within a given cell type was first summarized into an expression profile, where single-cell count values were replaced by 10 percentile values (deciles) (Fig. 2b, "Methods"). Intuitively, this reduced number of values captures the behavior of that transcript in the cell type, as inferred given cell-level observations. Next, to grasp similarities between expression distributions across cell types, pairwise Pearson correlations were computed using percentile-summarized isoform expression, resulting in a more meaningful distribution of correlation values than obtained with traditional metrics (Fig. 2b). In this manner, we managed to extend the notion of cell-type markers to rely not only on mean or frequency of expression, but on their actual distributional pattern. Our co-expression metric, therefore, by-passes cell-level matching of individual observations, providing a correlation estimate that is both robust to the uncertainty of single cell

expression and interpretable as a measure of expression similarity. Remarkably, changing percentile number did not have a noticeable effect on the resulting correlation values, as long as it ranged between 100 and 4 (Supplementary Fig. 2c). Nevertheless, using 1 percentile (median expression), substantially disrupted the correlation value distribution (Supplementary Fig. 2d), stressing the importance of selecting a sufficient number of percentiles to avoid over-summarizing isoform expression.

To detect modules of co-expressed isoforms, we used percentile correlations as a distance metric for hierarchical clustering, and designed a semi-automated cluster refinement approach to ensure maximal profile similarity within clustered modules (Fig. 2c, see "Methods" for a detailed description). First, the *dynamicTreeCut* R package[56] was used to initialize clustering. The dynamic clustering algorithm enables the selection of adaptive thresholds for better detection of clusters within the dendrogram. 166 clusters were obtained as a result. These were then re-clustered to mitigate the presence of highly similar (i.e., redundant) expression profiles (Supplementary Fig. 3a). To achieve this, cluster *metatranscripts* (i.e., the mean scaled expression of all isoforms in the cluster, see "Methods") were computed and hierarchical clustering applied, generating 26 clusters. At this point, 2381 isoforms remained unclustered, including isoforms from 3 groups that presented noisy expression profiles (Supplementary Fig. 3b). These were assigned by maximizing the similarities between cluster and isoform expression profiles, i.e., using the percentile correlation between isoform expression and cluster metatranscripts (see "Methods"). Finally, clusters were merged again to obtain completely unique profiles. Of note, we spotted two pairs of clusters with clearly similar profiles that were not merged metatranscript clustering. To avoid detection of falsely coDIU genes in downstream analysis, these were merged manually. As a result, we generated a total of 15 distinct clusters (Fig. 2d) containing all initially analyzed isoforms (6794 total) and representing diverse expression modalities across the 7 broad cell types. The different steps in the clustering process are implemented as functions in the *acorde* R package.

**Validation of percentile correlations on simulated data.** Building on studies reporting the poor performance of popular correlation metrics in single-cell data, authors have attempted the implementation of sparsity-aware measurements[52,57] and reported the potential of other alternatives to compute similarity, such as proportionality metrics[42]. Here, we present an interpretable, scalable and biology-aware alternative to single-cell co-expression studies based on Pearson or Spearman correlation. However, to better understand how percentile correlation performs in comparison to extant correlation metrics, we compared it to Pearson, Spearman and zero-inflated Kendall correlations[57] and one proportionality metric, rho ($\rho$)[58] using simulated data.

Given that there are -to the best of our knowledge- no scRNA-seq data simulators that include transcript co-expression patterns, we designed a simulation strategy (Supplementary Fig. 4, "Methods") to generate an appropriate validation framework for our metric. Briefly, we applied *SymSim*[59] to simulate a single-cell RNA-seq dataset (8 cell types, 1000 cells and 8000 transcripts, Supplementary Fig. 5a) and used the simulated expression values to artificially create 3000 synthetic transcripts showing 15 different expression profiles across the 8 cell types (Supplementary Fig. 5b). As a result, our simulated dataset contained 15 simulated clusters with distinct expression profiles while preserving the original cell type structure generated by *SymSim* (Supplementary Fig. 5a). Among them, clusters 1–5, 6–10 and 11–15 included transcripts showing high expression in one, two and three cell types, respectively, gradually increasing simulated

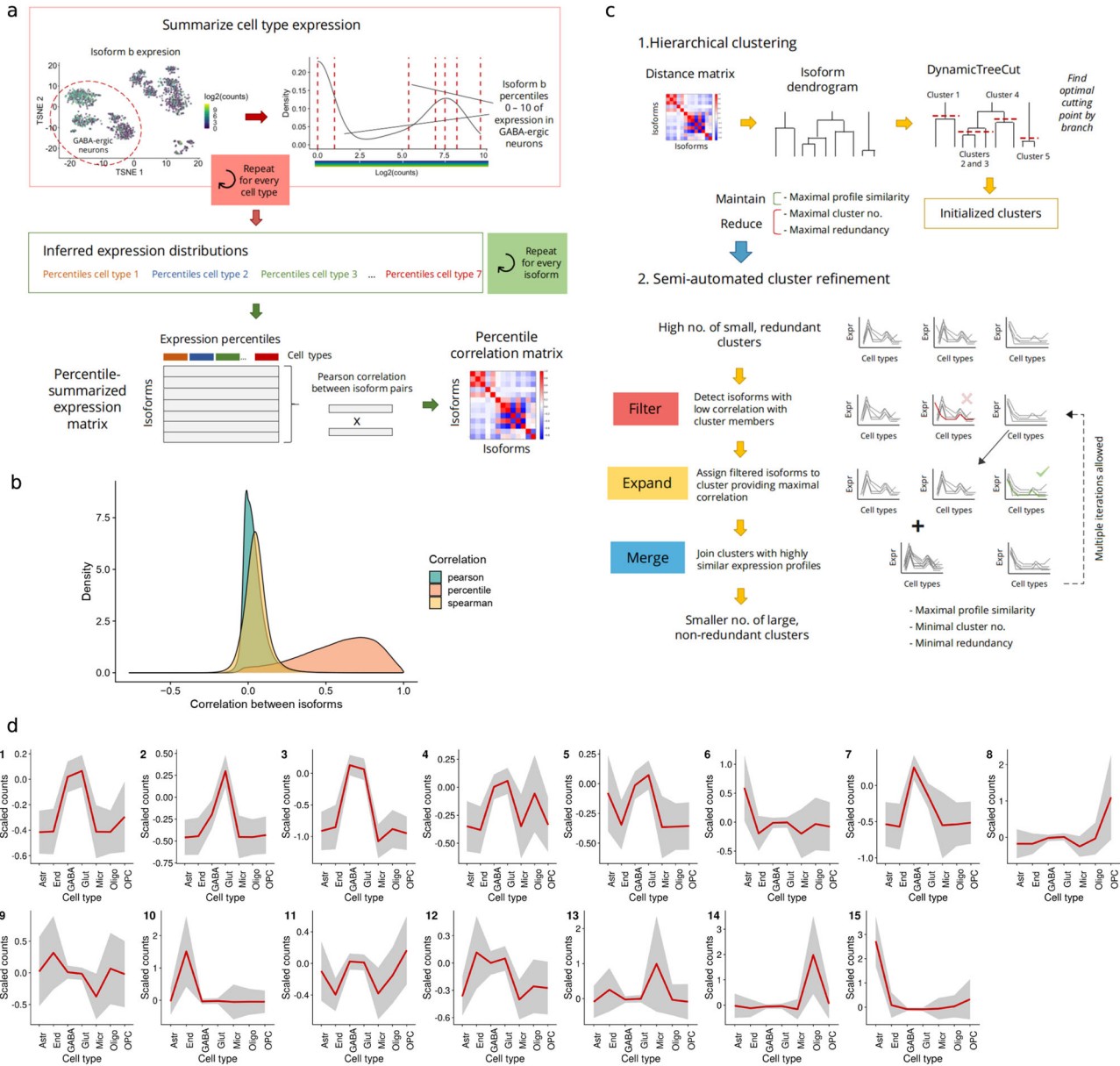

**Fig. 2 Percentile correlations and isoform clustering. a** Percentile correlation algorithm. For each isoform, cell type-level expression is summarized using percentiles (0–10) as a proxy of the isoform's expression distribution in each of the cell types. Then, Pearson correlations are computed using the percentile-summarized expression of all isoforms, obtaining a percentile correlation matrix. **b** Correlation density distributions. Pairwise isoform correlations were computed using Pearson, Spearman, and percentile+Pearson correlation. **c** Clustering pipeline. The percentile correlation matrix is first used as a distance matrix for hierarchical clustering. After dynamic cluster generation, noisy clusters are refined by a three-step semi-automated process. **d** Clusters generated after applying the *acorde* clustering pipeline to the mouse neural dataset. Cell-level mean expression (scaled, see "Methods") is computed for all transcripts and then aggregated as the global cell type mean, represented by the red line. Gray area corresponds to cell type mean ± standard deviation. Astr: astrocytes, End: endothelial cells, GABA: GABA-ergic neurons, Glut: glutamatergic neurons, Micr: microglia, Oligo: oligodendrocytes, OPC: oligodendrocyte precursor cells.

pattern complexity (Supplementary Fig. 5b, c). After refinement (Supplementary Fig. 5c, "Methods"), 1790 synthetic transcripts remained distributed across the 15 simulated clusters in groups ranging from 180 to 60 transcripts (Supplementary Fig. 5d).

In order to evaluate how well the 5 correlation methods recapitulated the simulated patterns, we computed these metrics for all synthetic transcript pairs in each simulated cluster (Supplementary Fig. 5e). Among them, percentile correlation consistently yielded the best proportion of high within-cluster correlations followed by $\rho$. However, rather counter-intuitively, $\rho$ had only an average performance when low-complexity patterns were provided,

with less than 20% output proportionality values >0.8 within clusters 1–5. Shockingly, zero-inflated Kendall correlation, a single cell-tailored metric, failed to recapitulate the simulated co-expression profiles and showed a considerably lower proportion of high correlations within the simulated clusters than Pearson and Spearman correlations. To better assess the ability of each metric to discriminate true from spurious co-expression, pairwise correlations for isoforms within (intra-cluster) and between (inter-cluster) clusters were compared. Even though results showed overall good separation between pairs from the same cluster, percentile correlation was the only metric to provide a complete lack of

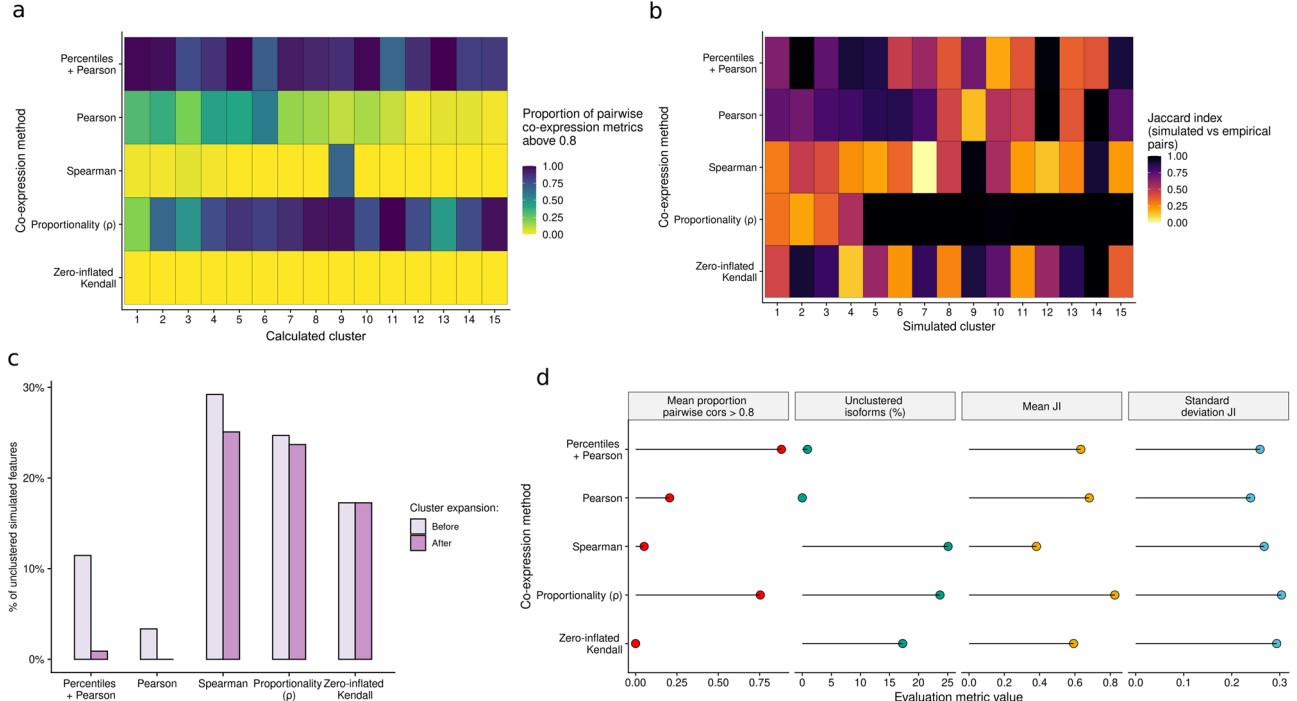

**Fig. 3 Comparative evaluation of percentile correlation on simulated clusters. a** Proportion of co-expression values above 0.8 in each empirical cluster, obtained after running the *acorde* clustering pipeline on simulated clusters using several correlation and proportionality methods as a distance metric. **b** Jaccard index of simulated vs calculated clusters obtained with each evaluated co-expression method. Simulated clusters were paired with one calculated cluster based on mean profile similarity, and synthetic transcripts in each of the paired clusters were compared. **c** Percentage of unclustered isoforms generated by each co-expression method. Results are shown before and after re-assigning unclustered isoforms by co-expression with the mean profile of extant clusters (i.e., cluster expansion). **d** Evaluation metric overview. Metrics are specified in the grid headers. *x*-axis shows values of the different metrics, *y*-axis displays evaluated co-expression methods.

overlap between inter and intra-cluster correlation distributions (Supplementary Fig. 5f). As a result of this evaluation, we can confidently assume that percentile correlations are useful to detect co-variation patterns, yielding overall higher correlation values than all other considered metrics (Supplementary Fig. 5E) and correctly discriminating true correlated and uncorrelated transcript pairs (Supplementary Fig. 5f). Of note, and similarly to real data, the simulated dataset showed no detectable effect when varying the number of percentiles used to compute percentile correlations (Supplementary Fig. 6).

Next, we compared the ability of each co-expression metric to inform clustering and group transcripts with similar expression patterns. To achieve this, we run our clustering pipeline on the simulated isoforms using the 5 metrics as distance (Supplementary Figs. 7–11). To enable benchmarking, clustering was automated to generate a total number of 15 calculated clusters (see "Methods"). In order to evaluate which metric worked best to detect co-expressed transcript groups, we considered internal correlations between the transcripts in the calculated clusters. We observed that $\rho$ and percentile-generated clusters, unlike the remaining co-expression metrics, presented consistently high levels of internal correlation (Fig. 3a). Notably, the distribution of correlation values obtained using percentiles was the most robust among the five metrics (Supplementary Fig. 12). We next assessed how well the clusters generated using each correlation metric (i.e., calculated clusters) recapitulated the simulated clusters. Calculated and simulated clusters were paired based on the similarities between their mean cluster profiles (Supplementary Fig. 13, "Methods"), and the Jaccard index was computed for each simulated-calculated pair to measure the agreement in transcript assignment (Fig. 3b). Interestingly, results were highly

heterogeneous for most methods: even though a number of simulated co-expression groups were easily detected by most metrics, no method was able to fully recapitulate the simulated clusters, with $\rho$ proportionality, Pearson, and percentile correlations being the most accurate (Fig. 4b). Zero-inflated Kendall and Spearman correlations, on the other hand, showed consistently low agreement with the simulated transcript groups. Finally, we considered the number of transcripts that remained unclustered (Fig. 3c) before and after re-clustering unassigned transcripts (cluster expansion step, see "Methods"). Pearson correlation provided successful cluster assignment for practically all transcripts in the simulated dataset, especially when incorporating percentiles (Pearson: ~10% unclustered before expansion, ~1% after; percentile: ~4% unclustered before expansion, 0% after), whilst the rest of metrics performed significantly worse, leaving 20–30% of transcripts unassigned even after cluster expansion, with proportionality (~30% unclustered before expansion, ~25% after) being the less optimal. Altogether, even though $\rho$ demonstrated good performance in many aspects of clustering, including intra-cluster correlation and agreement with the simulated clustering, it was outperformed by percentile correlation when globally considering all evaluated parameters (Fig. 3d). In addition to the fact that $\rho$ failed to control for unassigned transcripts, computing means and standard deviations of Jaccard indices across simulated-calculated pairs showed percentile and Pearson correlations as the most consistently accurate methods. All in all, our synthetic data evaluations showed that the percentile correlation approach performed well—and more consistently than $\rho$ proportionality- in all the evaluated features, and visibly captured co-expression better than both traditional and zero inflation-aware correlation metrics.

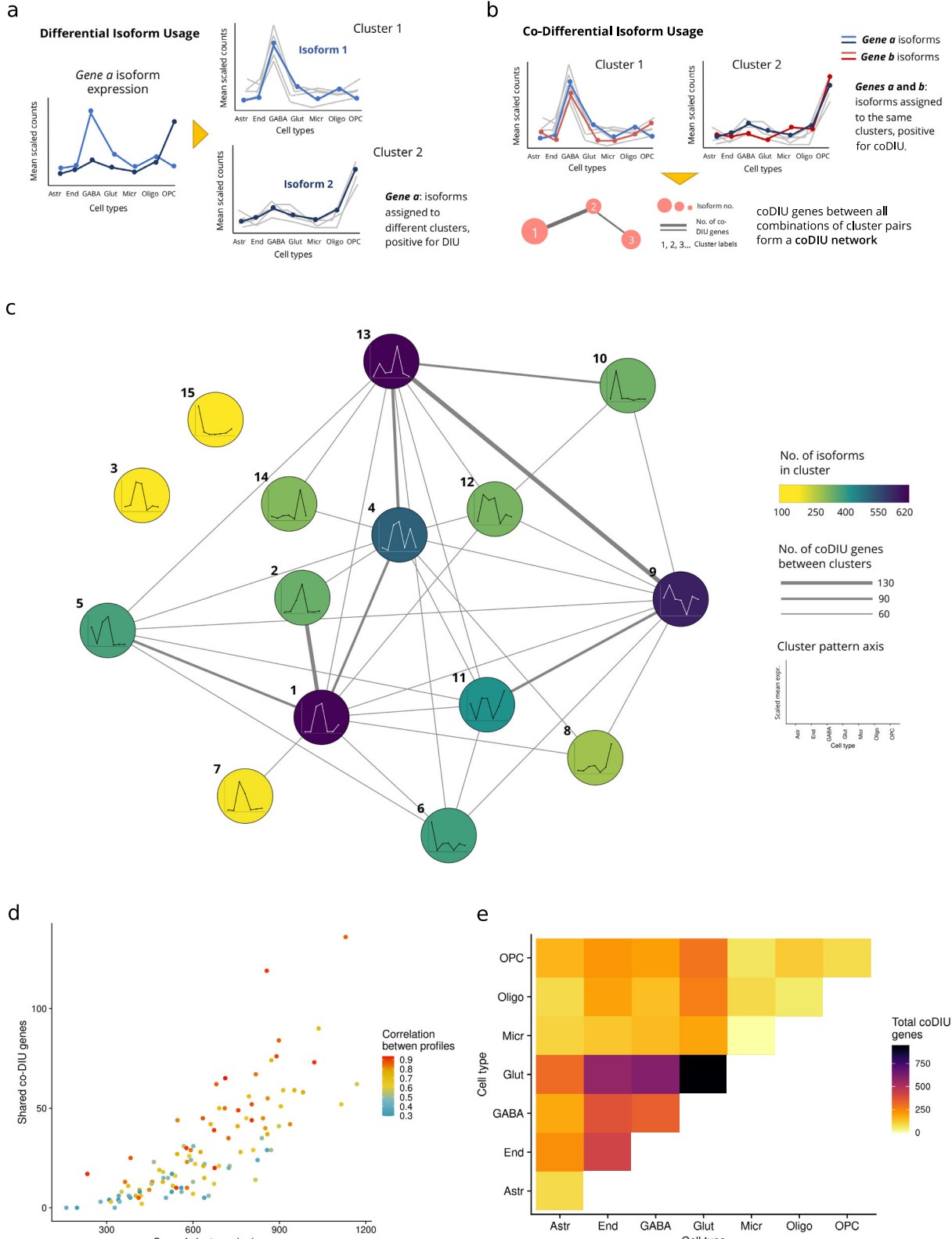

**Co-Differential Isoform Usage analysis of single-cell isoform expression.** Isoform clusters represent groups of alternative transcripts that are co-expressed at the cell type level. However, our clustering results did not provide information on iso-transcriptome properties associated with splicing regulation. To facilitate interpretation of the isoform clustering results, we first defined genes with DIU as those whose isoforms were assigned to different clusters (Fig. 4a). ~23% of total analyzed genes (2017 out of 8814) and 75% of genes with clustered isoforms (total = 2696) were positive for DIU, involving 5278 clustered isoforms. In this context, DIU genes will necessarily have two or more isoforms with significant changes in expression across cell types and simultaneously undergo cell type-dependent post-transcriptional regulation, leading to changes in isoform expression in each cell type.

**Fig. 4 Characterization of genes with co-Differential isoform usage. a** Cluster-based definition of Differential Isoform Usage (DIU) across multiple cell types. DIU genes have at least two isoforms assigned to different clusters, indicating a differential isoform selection pattern across the different cell types. **b** Definition of co-Differential Isoform Usage (coDIU) using clusters. CoDIU genes have multiple isoforms assigned to the same clusters, establishinig across-cell type co-expression relationships for at least two of their isoforms. **c** coDIU network. Nodes represent clusters and depict their mean expression profile across cell types. Node color represents cluster size (i.e., no. of isoforms in cluster). Edge width represents number of coDIU genes detected between each pair of clusters. coDIU genes are considered if they have at least one significant isoform co-expression pattern with one other gene. **d** Evaluation of cluster profile similarity and size as a function of the number of coDIU genes detected by *acorde*. X-axis corresponds to the sum of isoforms in each possible pair of clusters generated from the data. Y-axis contains the number of coDIU genes between the pair. Dot color represents the correlation between the mean expression profiles of each pair of clusters. The number of coDIU genes between a pair of clusters is seemingly related to the size of the clusters involved, and shows no relationship with the degree of similarity between the expression profiles of clustered isoforms. **e** Cell type-level coDIU patterns. For each pair of cell types represented in x and y-axis, heatmap color corresponds to the total number of genes found to be co-DIU between them. Total coDIU genes are calculated as the sum of coDIU genes detected between all cluster pairs that show high expression of isoforms in these cell types. GABA and Glut cell types share the highest number of coDIU genes, both with each other and with other cell types. Astr: astrocytes, End: endothelial cells, GABA: GABA-ergic neurons, Glut: glutamatergic neurons, Micr: microglia, Oligo: oligodendrocytes, OPC: oligodendrocyte precursor cells.

In order to study isoform co-expression patterns, we defined coDIU genes as those showing coordinated cell-type-specific isoform usage. Specifically, we considered two or more genes to be coDIU if their isoforms had been assigned to the same clusters (Fig. 4b, "Methods"). This resulted in the definition of an isoform co-splicing network, where nodes were clusters of correlated isoforms and edges represented coDIU genes, i.e., the number of genes for which two or more isoforms are co-expressed across cell types (Fig. 4c, "Methods"). To ensure the reliability of the detected coDIU patterns, a generalized linear model (GLM) was fitted for every pair of coDIU genes; selecting pairs that showed significant cluster-dependent expression variation across cell types and no significant changes in expression when only accounting for gene-level expression (see "Methods"). CoDIU genes therefore present cell type-dependent co-expression of at least two isoforms, represented by cluster assignment matches, but are not co-expressed when only gene expression is considered. Using this strategy, 1784 genes with at least one significant coDIU partner (*cluster\*cell-type* FDR < 0.05, *gene\*cell-type* FDR > 0.05) were detected, involving 5274 co-expressed isoforms. The number of coDIU genes sharing isoforms across each cluster pair was variable, although it rose up to >130 for highly connected clusters (Fig. 4c).

We then interrogated the coDIU network to find patterns underlying the splicing coordination signal detected by the *acorde* pipeline. First, in order to measure whether coDIU generated strong or subtle variations in isoform selection across cell types, we investigated the association of coDIU to single or multiple cell type isoform switching events. Note that single isoform switching events involve clusters with patterns that are similar across all cell types except one, leading to high between-cluster correlations. Interestingly, we found that the number of coDIU genes linking isoform co-expression clusters was dependent on cluster sizes, but showed no direct relationship with the similarities between expression profiles (Fig. 4d). The detection of coDIU genes involving isoforms with highly different expression patterns suggested that coordinated isoform usage is able to produce strong cell type-level shifts in isoform selection.

We next evaluated the cell type-level relationships in the isoform co-expression network, namely the occurrence of coDIU across all possible pairs of cell types in our data. Although co-splicing could potentially occur between any combination of cell types, results showed that a high proportion of coDIU interactions were detected when the isoforms involved had high expression in one of the two neural cell types, i.e., GABA-ergic and glutamatergic neurons (Fig. 4e). This can be partially explained by the fact that some of the clusters with neuron expression are among the largest generated by the *acorde* pipeline (Fig. 4c). However, another plausible explanation is that the central role of neurons in the tissue under study (i.e., primary visual cortex) might situate co-splicing at the core of neural function regulation, as well as the modulation of its interaction with glial cell types.

**Functional analysis of the coDIU network**. We next set out to investigate the functional implications of our isoform co-expression network. Since, with a few exceptions[60,61], splicing analysis tools rarely integrate functional information, we annotated the long read-defined transcripts using IsoAnnotLite (https://isoannot.tappas.org/isoannot-lite/). The resulting functional annotation included both transcript and protein-level motifs, sites, and domains, as well as non-positional, gene-level features such as Gene Ontology (GO) terms. A detailed description of the annotation process and a comprehensive list of functional categories and source databases are available in Supplementary Note.

First, we analyzed which biological processes and gene functions were potentially controlled by DIU (AS-regulated) and coDIU (co-regulated) mechanisms, that is, which gene functionalities were overrepresented in the DIU and coDIU sets. In order to discriminate the functional properties of AS-regulated genes from those showing no cell type specificity in isoform expression, we performed a functional enrichment test for DIU genes vs genes with DE isoforms, which were used as the background (Supplementary Fig. 14, see "Methods"). Interestingly, DIU genes showed significant enrichment (FDR < 0.05) in GO terms associated with gene expression regulation, including general mechanisms (nucleic acid metabolic process) and processes related to both DNA (DNA metabolic process) and RNA metabolism (RNA metabolism). In addition, genes annotated as participating in protein-complex mechanisms required for these processes (protein-containing complex) were found to be significantly overrepresented in the DIU group. Remarkably, functional enrichment revealed that DIU genes were enriched in binding sites for miR-412-3p. Even though extant literature includes no functional roles for this miRNA in the brain, miR-412-3p has been found to interact with Mbnl1-AS1[62], a long non-coding RNA that is also an antisense isoform of Mbnl1, which is an important splicing regulator in the neural context[63–65]. We speculate that differential inclusion of miR-412-3p binding sites in the isoforms of coDIU genes might be related to additional regulatory roles of this miRNA in neural cell types.

Next, in order to investigate the cellular processes where coDIU could potentially have a relevant regulatory role, we compared the proportion of coDIU and DIU genes annotated for each functional feature in the transcriptome using a partially-overlapping samples z test[66] (see "Methods"). In total, 91 positional functional features and 59 GO terms were found to

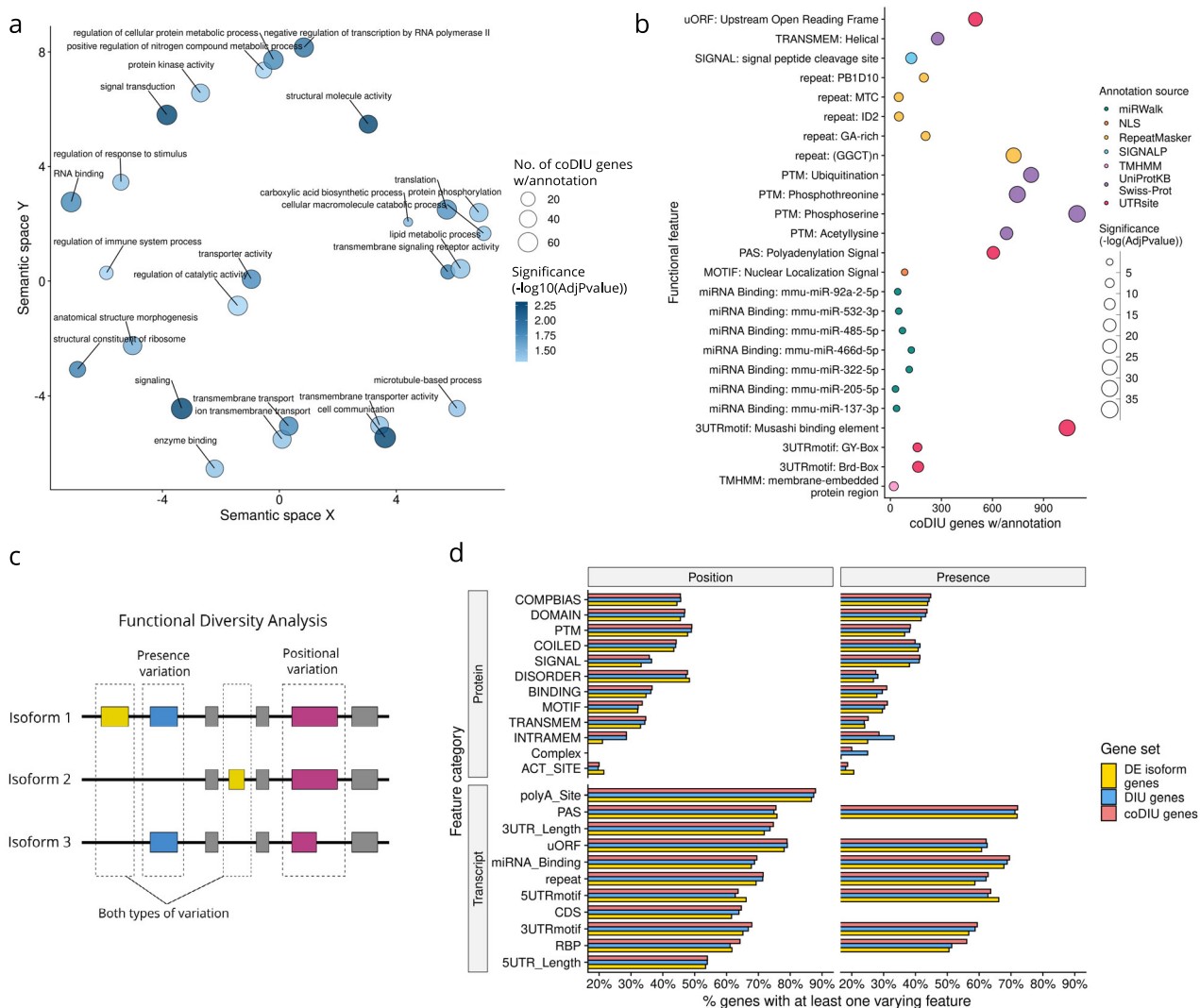

**Fig. 5 Functional analysis of DIU and coDIU genes. a** Gene Ontology (GO) Functional Enrichment results for co-Differential isoform Usage (coDIU) vs genes with Differential Isoform Usage (DIU) of isoforms. Only terms obtained after Revigo filtering are shown. *x* and *y*-axis indicate semantic similarity, as defined by Revigo (see "Methods"). Dot size represents total coDIU genes annotated for each GO term. Dot color represents significance, i.e., −log10(adjusted *p*-value). **b** Functional Enrichment of positional features for coDIU vs DIU genes. *x*-axis indicates the total number of coDIU genes including the tested annotation feature, *y*-axis shows functional features. Dot color represents the functional category (i.e., annotation source database) and dot size represents −log(adjusted *p*-value). **c** Schematic representation of tappAS' Functional Diversity Analysis (FDA). The genomic position and present/absence criteria, both of which can be used to detect functional variation among isoforms from the same gene (i.e., label the gene as *varying*), are depicted. **d** FDA results for DE isoform, DIU, and coDIU genes. *y*-axis shows transcript and protein functional categories (see Supplementary Note for category definition information). *x*-axis shows the percentage of genes including at least one feature annotation from each of the categories that are detected as functionally *varying*. Both FDA criteria are shown (position: left grid column, presence: right grid column). CoDIU genes show the largest level of functional variation.

be annotated in a significantly higher proportion in coDIU vs DIU genes (FDR < 0.05, Supplementary Data 1). Given the extensive list of significant features obtained, we focused on the most relevant set of features for visualization and interpretation purposes (Fig. 5a, b). Full results, however, are available in Supplementary Data 1. First, to remove redundant functionalities, GO terms were filtered by semantic similarity using Revigo[67] (see "Methods", Fig. 5a), resulting in 26 unique terms. Among them, and similarly to DIU genes, coDIU genes showed significant enrichment (FDR < 0.05) in functionalities related to specific aspects of transcriptional regulation (regulation of nitrogen compound metabolic process, regulation of transcription by RNA polymerase II, Fig. 5a, Supplementary Data 1). However, genes that were positive for coDIU were also significantly

associated with signaling mechanisms (protein kinase activity, protein phosphorylation, signal transduction), membrane transport and cell-to-cell communication (transmembrane signaling receptor activity, cell communication, regulation of response to stimulus), and metabolism (carboxylic acid biosynthetic process, cellular macromolecule catabolic process, lipid metabolic process). Remarkably, coDIU genes showed additional enrichment for post-trancriptional processes and functionalities such as RNA binding and translation. This result links genes participating in RNA metabolism with the coordination of AS, and suggests that the co-expression of alternative isoforms may contribute to the fine-tuning of post-transcriptional regulatory processes. Regarding positional functional features, coDIU genes presented a significantly higher proportion of miRNA binding, 3′UTR and

upstream Open Reading Frame (uORF) motifs (FDR < 0.05, Fig. 5b) as well as several predicted protein elements such as post-translational modifications (PTMs), nuclear localization signals (NLS) and transmembrane domains. These motifs and domains exert a broad variety of roles, including interactions with regulators (e.g., 3′UTR motifs), mRNA turnover control (e.g., uORFs), and protein-specific localization (e.g., NLS, transmembrane), indicating the potential of coDIU to create functional synergies via the co-inclusion of specific domains and motifs in an isoform-specific manner.

To further explore the potential of domain-including (or excluding) isoform co-expression across neural cell types, we performed a Functional Diversity Analysis (FDA, Fig. 5c). FDA is part of the tappAS framework[61], and identifies functionally *varying* genes, i.e., genes expressing transcript variants with differences in the inclusion of functional features (see "Methods"). FDA can be evaluated from a presence/absence standpoint (i.e., AS completely removes a feature), or by detecting variation in the transcriptomic positions defining the feature (Fig. 5c). We, therefore, compared the diversity in transcript-level functional features between DIU, coDIU genes, and genes with more than one DE isoform, for all functional categories provided by IsoAnnotLite (Fig. 5d). Interestingly, we observed that the percentage of *varying* genes increased with isoform expression complexity, with coDIU resulting in the largest amount of feature inclusion diversity in virtually all protein and transcript feature categories. To measure this effect, we compared the percentages of variation for all pairwise combinations of the three gene sets (paired samples t-test, see "Methods") and confirmed the observed trend, regardless of whether the variability criteria employed (position/presence). In particular, even though nearly all comparisons were significant, coDIU resulted in the most significant increase in feature variation (coDIU vs DE isoform genes p-value: presence = 1.14e−03, position = 8.92e−04; coDIU vs DIU genes p-value: presence = NS, position = 2.04e−03). We verified that this increase in functional diversity was not associated to expression level or isoform length biases in coDIU genes (Supplementary Fig. 15a, b). This result suggests that alternative isoforms that engage in co-expression relationships tend to alter their functional properties significantly more often than other transcripts, namely by changing the inclusion of motifs, sites and/or domains, thereby coupling AS and isoform co-expression with functional potential.

**Functional analysis of neuron-oligodendrocyte isoform co-expression**. To further understand the relationship between cell-type identity, isoform co-expression, and the functional properties of coDIU genes, we searched the coDIU network for cluster groups representing biologically-related isoform switches between neural cell types. Namely, we focused on a set of 118 coDIU genes (Fig. 4c) containing isoforms with higher relative expression in oligodendrocytes (cluster 14), neurons (GABA and Glutamatergic neuron cell types, cluster 1) or both (cluster 4, Fig. 6a) and analyzed isoform-associated functional variability using FDA (Fig. 6b). For this set of alternative isoforms, 3′UTR length showed the highest variation rate among annotated transcript-level functional categories (*varying* in ~70% genes, Fig. 6b). Moreover, we noticed that these changes followed a clear cell type-specific pattern, with the majority of coDIU genes showing higher relative expression of their longest 3′UTR isoforms in neurons (Supplementary Fig. 16a) and neural-specific isoforms generally expressing longer 3′UTRs than their oligodendrocyte-expressed counterparts (Fig. 6c).

Next, we inspected several 3′UTR-related functional categories (repeat regions, miRNA binding, and 3′UTR motifs) using ID-level

FDA (see "Methods", Supplementary Fig. 16b) to identify specific functional features associated to isoform usage differences between neurons and oligodendrocytes. Regarding the presence of miRNA binding sites, in spite high varyiation rates, no specific miRNA motif was shared by more than ~10% of genes (maximum of 12 out of 118 genes for miR-495-3p), while some repeats, such as (GT)n, were present in ~25% of coDIU genes across the three clusters with varying rates >50%. Nevertheless, in the case of 3′UTR motifs, we found that Musashi binding motifs presented inclusion changes in ~60% of annotated coDIU genes (Supplementary Fig. 16b). The Musashi protein is known to be a neural RNA-binding protein that participates in translational control, regulating cell fate and cell cycle[68,69]. In line with this, the coDIU network included several genes in which 3′UTR elongation led to neuron-specific co-inclusion of Mushashi binding elements, including kinase-encoding genes *Ppip5k1* (diphosphoinositol pentakisphosphate kinase 1, Supplementary Fig. 17a) and *Prkcz* (protein kinase C zeta type, Supplementary Fig. 17b). These results align with the previously shown enrichment of signaling and translation-related genes within the coDIU network (Fig. 5a) and hints that co-expression of Musashi-binding isoforms may generate 3′ UTR binding-mediated changes in the translation of proteins participating in different signaling pathways.

Importantly, the majority of neuron-oligodendrocyte coDIU genes also presented frequent coding region variation (i.e., CDS, Fig. 6b), revealing that coordinated isoform usage can modify both transcript and protein functional properties. In particular, protein domains (PFAM) and post-translational modifications (PTMs) presented high variation rates (*varying* in ~30% of genes containing the feature, Fig. 6b) and thus constituted the categories with the most cell type-dependent functional variation. While ID-FDA reported no specific PFAM domains shared among the analyzed gene set (~1%, maximum of 2 out of 118), up to ~10% of them presented inclusion variation in similar PTMs (12 out of 118) with medium to high variation rates for phosphorylation, acetylation and ubiquitination (Supplementary Fig. 16b). However, synergies between PTM and domain inclusion changes could still result in differential functional activities at the protein level. As an example, we found two genes involved in different aspects of RNA metabolism, *Lrif1* (ligand-dependent nuclear receptor interacting factor, Fig. 6d) and *Stau2* (Staufen RNA binding protein homolog, Fig. 6e), both of which present cell type-level domain inclusion associated to coordinated changes in the expression of alternative protein isoforms. In humans, *Lrif1* has been shown to interact with a number of nuclear receptors, including retinoic acid receptors, to suppress the ligand-mediated transcriptional activator role of these proteins[70]. This interaction occurs at the N-terminal end, which presents differential inclusion of several protein motifs and domains among *Lrif1* isoforms. Specifically, two *Lrif1* isoforms that are depleted in oligodendrocytes present inclusion of a coiled/disordered region as well as binding and phosphorylation sites (Fig. 6d), which may be connected to a specific transcriptional regulation role in mouse neuron cells. Also regarding the neural-related functionality of these genes, the rat homolog of *Stau2* is known to have a role in mRNA transport from the nucleus to neuron dendrites[71]. In our system, *Stau2* isoforms upregulated in oligodendrocytes show a C-terminal Staufen domain that is not included in neural-specific isoforms (Fig. 6e) and is responsible for Staufen dimerization in humans[72]. On the other hand, one of the neuron-expressed isoforms includes an extra N-terminal RNA binding domain. Differences among *Stau2* isoforms may be connected to a dual role for this protein in neurons and glia in which enhanced RNA binding activity is required in neurons, while Staufen dimers may be more likely to form in oligodendrocytes. However, further analyses and validation experiments are required to confirm these

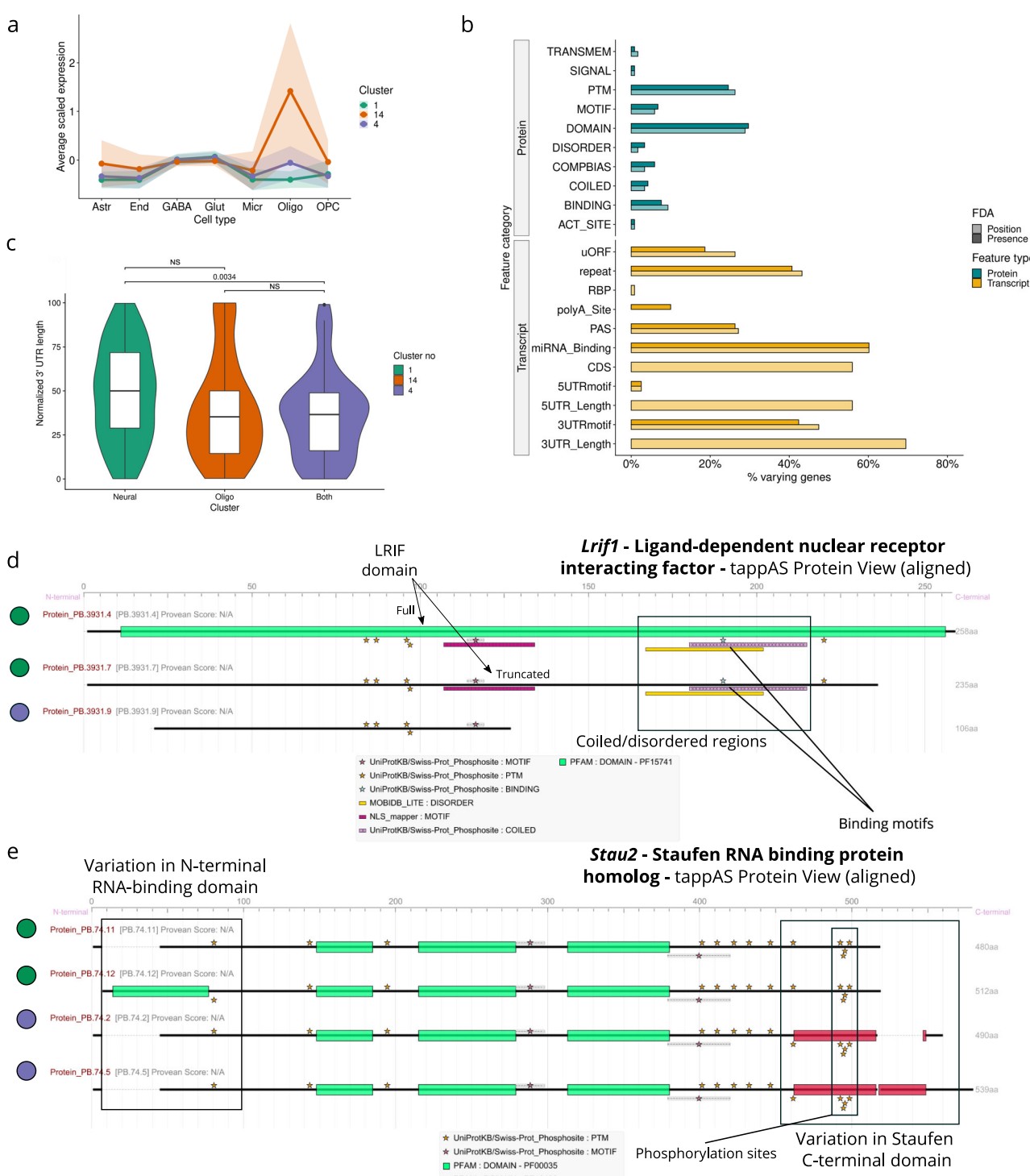

**Fig. 6 Functional analysis of the 118 coDIU genes detected across neural-oligodendrocyte clusters (no. 1, 4, and 14). a** Cell type expression patterns of clusters selected for downstream functional analysis: neural (cluster 1, green), oligodendrocyte (cluster 14, orange) or shared (cluster 4, purple). For each cluster, cell-level mean expression (scaled) is computed for all transcripts, and cell means are aggregated to obtain a global cell type mean, represented by the lines. Colored areas correspond to cell type mean ± standard deviation. **b** Functional Diversity Analysis (FDA) results. *y*-axis: functional annotation categories. *x*-axis: percentage of genes including at least one *varying* feature from a given functional category. **c** Violin and boxplots of normalized 3′UTR lengths for isoforms in each of the three neural-oligodendrocyte clusters (*n* = 177 transcript isoforms from coDIU genes with 3′UTR variability). Normalized lengths are computed by dividing each individual isoform's 3′UTR length by the sum of 3′UTR lengths of all the gene's isoforms. Violin plots indicate density distributions. For boxplots, boxes indicate median (middle line), 25th (Q1) and 75th (Q3) percentiles (box hinges); whiskers represent min = Q1 − 1.5 · Interquartile Range (IQR) and max = Q3 + 1.5 · IQR; dots constitute outliers. Significance levels for the comparison of the three groups are indicated above the corresponding braces (*p*-value, Wilcoxon's test, two-sided). NS: not significant. tappAS view of **d** *Lrif1* and **e** *Stau2* protein annotations. Cluster assignments for each isoform are indicated by dot color.

findings and reveal additional coordination of domain inclusion changes for genes involved in differential RNA processing between neurons and glial types.

*Analysis of coDIU in GABA-ergic neuron subtypes.* In order to showcase the applicability of the *acorde* pipeline and test its performance under high-granularity conditions, we analyzed isoform expression in the primary visual cortex among 5 GABA-ergic neuron subtypes (Lamp5, Pvalb, Sncg, Sst, Vip, 4921 total cells post-QC) defined in another recent study by Tasic et al.[44] (hereby referred to as Tasic 2018 dataset). Remarkably, quantification of isoform expression using the previously-defined long read transcriptome (see Supplementary Note) resulted in no large transcriptomic differences among the subtypes, as revealed by UMAP[73] dimension reduction (Supplementary Fig. 18a), with Pvalb neurons showing the largest differences with the rest of the cell types. Accordingly, DE analysis between the five groups only returned 568 significantly DE isoforms (FDR < 0.05, fold-change > 1.5, see "Methods"), which make up only ~4% of all isoforms included in the analysis (13870 isoforms remaining post-QC). Using percentile correlations, DE isoforms were grouped in 171 small clusters, which were then merged into 5 distinct expression profiles (Supplementary Fig. 18b) using dynamic hierarchical clustering (see "Methods"). The largest among the 5 clusters contained those isoforms with higher relative expression in Pvalb neurons (cluster 2, 184 isoforms, Supplementary Fig. 18b), in agreement with the global data structure (Supplementary Fig. 18a).

DIU and coDIU relationships encoded by these co-expressed isoform modules were next investigated. We found 22 DIU genes among the 5 clusters, 16 of which presented significant coDIU relationships with at least one other gene (see "Methods"), involving 36 isoforms in total. In line with the clustering and UMAP results (Supplementary Fig. 18a, b), most isoforms participating in coDIU relationships presented higher relative expression in Pvalb neurons (Supplementary Fig. 18c). We, therefore, decided to explore the functional features that varied among coDIU genes that had isoforms in the Pvalb expression cluster. Remarkably, two of the detected *coDIU* genes were membrane-associated proteoglycans *Gpc1* and *Tmeff2*, which have been described to have neuronal function in previous studies[74,75]. These genes presented coordinated switches in isoform expression in the Pvalb (cluster 1) and Scng (cluster2) subtypes (Supplementary Fig. 18d) as well as cell type-specific changes in the functional properties of their isoforms. Namely, *Gpc1* presented Pvalb-specific increase in expression of a signal peptide-including isoform (Supplementary Fig. 18e). Meanwhile, for *Tmeff2*, Pvalb-expressed isoforms were missing the C-terminal end, which included the transmembrane domain of the protein and several glycosylation sites (Supplementary Fig. 18f). Even though these -and other similar results shown in this manuscript- would require further validation, they serve to illustrate the potential of *acorde* to uncover candidates for functionally-relevant isoform co-expression relationships.

Of note, the low number of clusters and DIU/coDIU genes detected by the *acorde* pipeline in this dataset, explained by the high isoform expression homogeneity among GABA cell subtypes, precluded the generation of more comprehensive functional results. Even so, we strongly believe that all in all, these analyses demonstrate that *acorde* can be used to study isoform co-expression even in untoward scenarios, making a case for its usability in datasets with less well-defined cell types or lower signal-to-noise ratio.

## Discussion
AS is known to be a tightly-regulated process in which splicing factors interact to create cell-type-specific isoform expression

patterns[76]. The transcriptome-level consequences of AS regulation have been studied in different ways, including, but not limited to, the detection of within-isoform coordination of alternative sites[40,41], the generation of gene-isoform networks to uncover regulatory relationships[77–80] and the application of single-cell data to unravel cell type-specific expression patterns for same-gene isoforms[27,81]. However, the extent to which AS regulation creates co-expression patterns among alternative isoforms from different genes has not yet been fully addressed. Specifically, previous studies tackling this type of isoform co-expression have either focused on specific event types, such as alternative 3′ exons[82], or solely on the identification of functionally-relevant alternative isoforms in different biological contexts[83,84].

In this study, we presented *acorde*, an end-to-end pipeline to generate isoform co-expression networks and detect genes with coDIU, and applied it to the study of isoform co-expression among seven neural broad cell types[43] and five GABA-ergic neuron subtypes from an independent dataset[44]. To this end, we leveraged single-cell data by implementing percentile correlations, a metric designed to overcome single-cell noise and sparsity and provide high-confidence estimates of isoform-to-isoform correlation. Here, we show that percentile-summarized Pearson correlations outperform both classic and single-cell specific correlation strategies[57], including proportionality methods that were recently proposed as one of the best alternatives to measure co-expression in single-cell data[42]. In addition, using a long read-defined, functionally annotated transcriptome enabled us to obtain a biological readout from the isoform network. coDIU genes were found to be enriched in the same biological functions, a number of which were unique in comparison to enrichment results for genes solely reported as DIU. Inter-gene isoform co-expression thus appears to impact a subset of DIU genes sharing specific sets of functions, which suggests that coDIU may contribute with an additional layer of complexity to some cellular processes, operating as a fine-tuning mechanism. Our analyses also revealed that isoforms from coDIU genes encompass higher functional diversity than those belonging to DIU genes, an effect that was not associated to expression or length differences. Importantly, these changes impact both transcript and protein isoform functional features, which pinpoints their ability to globally increase the functional repertoire of coDIU genes. Mechanisms generating isoform co-expression can therefore be thought of as a potential source of functional synergies between alternatively spliced genes, giving rise to simultaneous changes in functional properties among co-expressed isoforms.

To demonstrate the power of the *acorde* pipeline, we include examples from both neural datasets where these kinds of coordinated changes were detected. First, we report a neural-specific pattern of 3′UTR co-elongation that is consistent with the available literature[85,86] and results in a simultaneous increase in the number of repeats, Musashi binding elements, and miRNA binding sites in these UTRs. In addition, we describe cell-type-specific co-expression of kinase isoforms that creates divergent functional properties among them, which points to a splicing-mediated coordination of signaling pathways. On a broader note, an interesting finding stemming from our functional analyses concerns the diversity of biological properties that make up the functional signal of the coDIU network. Namely, we found coDIU genes to be involved in a wide array of biological processes and showed how joint changes in isoform expression across cell types can result in the differential inclusion of features from a wide variety of functional categories. These diverse, complex results suggest that AS and post-transcriptional regulation mechanisms can modify the presence of a large variety of functional features in co-expressed isoforms, which may potentially

contribute to modulate key cell-level processes encoding cell-type identity. While these insights need to be subject to further experimental validation, they serve to illustrate the hypothesis-generating power of our pipeline.

All in all, we have hereby shown that *acorde* can effectively be used to leverage single-cell RNA-seq data to build isoform co-expression networks, providing tools for the exploration of the post-transcriptional regulation of gene expression from an innovative standpoint, including the disclosure of the cellular processes and functional properties impacted by these mechanisms.

## Methods

**Single-cell data pre-processing and quality control**. Mouse neural single-cell RNA-Seq data from mouse primary visual cortex was obtained from Tasic et al.[43] and consists in paired-end Illumina reads generated with the Smart-seq2 protocol[87], which enables isoform-level quantification. Reads were downloaded from Sequence Read Archive accession SRP061902 (https://www.ncbi.nlm.nih.gov/sra?term=SRP061902) and mapped to the mouse genome (GRCm38.p6) using STAR[88]. We performed isoform expression quantification of the long read-defined isoforms (see Supplementary Note for details on long read transcriptome definition) using RSEM[89], and used the labels provided by Tasic et al. to assign the 1679 cells to 7 broad cell types: 5 glial (microglia, endothelial cells, oligodendrocytes, oligodendrocyte precursor cells (OPCs) and astrocytes) and 2 neural (GABA-ergic and glutamatergic neurons).

Isoform length effect on expression was evaluated using the *NOISeq* R package[90], where mean expression showed to be highly correlated with transcript length (adjusted $R^2 = 0.81$; $p$-value $= 2.2e-16$). Using isoform $i$ effective length ($l_i$) and cell-level $j$ estimated counts ($c_{ij}$), both output by RSEM and, after testing several alternatives, we devised a custom formula to minimize the impact of length on isoform expression for each isoform $i$:

$$y_{ij} = \frac{c_{ij}}{\left(10^{-6}\sum_{i=1}^{I} c_{ij}\right)\sqrt{10^{-3} l_i}} \quad (1)$$

The transformed expression value for isoform $i$ in cell $j$ ($y_{ij}$) was again tested for length bias and a low correlation was found (adjusted $R^2 = 0.25$; $p$-value $= 6.84e-8$). Next, we inspected the library size distribution and filtered both high and low-count outliers due to potential premature cell death or library preparation duplets, with a total of 1591 cells passing quality control. Feature-level quality control was performed in a cell type-aware manner, keeping isoforms that showed non-zero expression in at least 25% of one cell type. Out of the 36,986 isoforms and 12,692 genes in the PacBio-defined transcriptome, we retained 16,240 isoforms and 8814 genes for downstream analysis.

### Differential expression (DE) across multiple groups

*Single-cell DE analysis.* DE analysis among the 7 cell types was performed by combining ZinBWaVE weights[48] and bulk-designed DE methods edgeR[50] and DESeq2[49] (i.e., using the corresponding R packages), which enable multiple group testing and were among the best-performing methods when combined with the ZinBWaVE method. Briefly, ZinBWaVE calculates cell-level weights for each isoform, effectively downweighting zeros during modeling for DE in single cell data (see van den Berge et al.[48] for details), and hence unlocking bulk RNA-Seq computational methods for single-cell data. Of note, GLM within edgeR and DESeq2 were built and run following the pipeline used by van den Berge et al.[48] to make them suitable for single-cell RNA-seq data, and are implemented in a wrapper function within our R package as described in Eq. 2, where $y_{ij}$ is expression of isoform $i$ in cell $j$, $T_{kj}$ is a dummy variable which takes value 1 when cell $j$ is assigned to cell type $k$ ($k = 1,...,K$) and 0 otherwise, $\beta_{ki}$ are the regression coefficients for isoform $i$, $\varepsilon_{ij}$ represents the error term, and $h()$ is the link function of the GLM (natural logarithm in this case).

$$h\left(y_{ij}\right) = \beta_{0i} + \sum_{k=2}^{K}\beta_{ki}T_{kj} + \varepsilon_{ij} \quad (2)$$

DE was defined using a significance threshold of FDR < 0.05 when testing the significance of the model for each isoform $i$, that is, $H_0$: $\beta_{2i} = ... = \beta_{Ki}$. Isoforms considered DE were preserved for downstream analysis if detected by at least one of the two methods edgeR or DESeq2, since this indicates a change in expression for any of the cell types considered rather than a flat expression profile.

*Neural cell sampling strategy to select a consensus set of DE isoforms.* Prior to DE testing, and to balance sample sizes across cell types, we performed 50 independent runs of random neural cell sampling (without replacement) followed by zero-expression filtering (expression above zero for at least 25% of cells in at least one cell type to control zero abundance among iterations and avoid problems during GLM modeling) and DE testing with edgeR and DESeq2. Specifically, the two neural cell types (GABA-ergic neurons, $n = 729$; glutamatergic neurons,

$n = 711$ cells) were downsampled by randomly selecting 45 cells, keeping $N = 241$ cells for multi-group DE testing.

To measure the consistency of each method independently, we calculated the mean and standard deviation of the number of DE isoforms across all same-method sampling runs ($R = 50$). To check the level of within-method agreement, we next considered isoform IDs labeled as DE in each independent method run, and calculated the Jaccard Index ($J_{rs}$) between DE results of that same method for all possible pairs of random sampling runs $r$ and $s$ ($r,s = 1,...,50$, $r < s$, a total of 1225 comparisons). To summarize this information, we relied on the mean and standard deviation of these two sets of $J_{rs}$ values. Finally, we measured the level of agreement between edgeR and DESeq2 regarding our DE criteria, that is, considering isoforms detected by at least one of the methods to be significantly DE (FDR < 0.05). To achieve this, we calculated the union of DE isoforms between one-to-one pairs of edgeR and DESeq2 runs ($R = 50$), and computed the Jaccard Index between all possible pairwise combination of global DE results, i.e., isoforms detected by at least one method (again, 1225 comparisons).

Using the 50 independent set of DE results, including both edgeR and DESeq2, we set out to define a consensus set of DE isoforms. To maximize sensitivity, we first considered the union of DE isoforms obtained by edgeR and DESeq2 on each of the downsampled versions of the data ($R = 50$), that is, isoforms detected to be significantly DE (FDR < 0.05) by at least one of the applied methods. Among the 50 lists of DE isoforms, those that were significant in at least 50% of the runs were selected. In addition, minor isoforms, i.e., those accumulating less than a 0.1 proportion of the absolute expression of their gene, were filtered. Finally, we retained only isoforms from genes with more than one DE isoform, hence removing cases where no AS-directed, isoform-level co-expression relationships can be established. This sets a general requirement for the entire study, which is that all isoforms retained must have, at all times, at least one same-gene counterpart to establish regulatory relationships that can be based on differential splicing of that gene, given that no AS regulation can be detected if a gene's total expression is represented by a single isoform.

**Percentile correlation**. In order to assess the similarity of isoform expression profiles across cells, a correlation measurement can be used by taking cells as observations. We propose here instead to first summarize the expression within a given cell type with percentiles and then compute the correlation using all cell types and their percentiles as observations, a method that we refer to as "percentile correlation" (see main text Fig. 2a).

Percentile correlations rely on the assumption that cell-to-cell differences can be mostly attributed to transcriptional stochasticity or technical noise, and that these within-cell type differences have a smaller effect than between-cell type expression differences. However, expression estimates for transcripts within the same cell are biased in different degrees, mostly depending on their expression levels, with lower expression being generally accompanied by higher noise levels[53]. This modifies the extent to which isoforms are affected by noise in each cell and causes strong cell-level effects that prevent the detection of co-expression relationships using solely cell-level measurements. Instead, we set out to target changes in expression across cell groups. We therefore considered isoform expression levels in the different cell types as a range of possible values, defined by the cell-level measurements in the data. In this context, the expression value of an isoform in a cell is used a proxy to infer the underlying distribution of expression values in the cell type, where the shape and width of this distribution will depend on both biological and technical factors.

To translate this into a metric, we first took the expression values of an isoform in each of the cell types and computed a number of percentiles ($p$). We selected $p = 10$ to achieve a good balance between accuracy and computational burden in downstream analysis. As the minimum expression value (percentile 0) was also included, we obtained 11 values representing the expression range within a given cell type. As a result, each isoform will possess a new, recalculated expression vector where the percentile values computed in each cell type will replace cell-level expression estimates. This process was repeated for each isoform. Next, we computed pairwise Pearson correlations between every pair of isoforms, obtaining a percentile correlation matrix **R**. In this context, high correlations will appear if a pair of isoforms shows a similarly broad expression distribution in most cell types, as well as a similar amount of relative expression change between cell types.

**Semi-automated isoform clustering**. In order to obtain modules of tightly co-expressed isoforms, we combined the hierarchical clustering algorithm with several rounds of cluster profile refinement (see main text, Fig. 2c), in order to automate the most intensive steps of clustering while also granting control over the level of aggregation and within-cluster similarity. Clustering and refinement steps can be combined and re-arranged to best capture co-expression patterns within the data, and their parameters can be defined by potential future users to provide maximum flexibility. Functions for clustering and refinement are implemented and documented in the *acorde* R package. Since the package vignette includes code-level details on how these functions work and how were used to generate the results in this manuscript, this section of methods will focus on describing the underlying processes in the package and how they were combined to obtain isoform clusters.

*Dynamic hierarchical clustering.* The previously obtained correlation matrix (**R**), where each element $r_{ij}$ represents the Pearson's correlation coefficient between the percentiles of isoforms ($i$,$j$), was transformed into a distance metric to be used in the hierarchical clustering. As we aimed to cluster positively correlated isoforms given our biological hypothesis, negative correlation values were discarded by replacing them with zero values, and therefore defined the distance between any pair of isoforms $i$ and $j$ as in Eq. 3.

$$d_{ij} = \begin{cases} 1 - r_{ij} & \texttt{if } r_{ij} > 0 \\ 1 & \texttt{if } r_{ij} \leq 0 \end{cases} \quad (3)$$

Hierarchical cluster analysis was performed using the *hclust()* function in the R *stats* package[91] with the average linkage criterion, and obtained a dendrogram. To obtain clusters, we used the *cutreeHybrid()* function in the *dynamicTreeCut* R package[56] in order to find different thresholds for different branches of the dendrogram tree, instead of using a fixed threshold for the entire dendrogram. The following non-default parameters were provided to the *cutreeHybrid()* function: *deepSplit = 4, pamStage = FALSE, minClusterSize = 20*. Briefly, *deepSplit* ranges between 0 and 4, and provides smaller clusters, more accurate clusters when set to high values. *pamStage* determines whether a second stage of clustering using an algorithm similar to the Partition Around Medoids (PAM) method will be performed after searching the dendrogram for clusters (see Langfelder et al.[56]). As a result of this PAM-like step, no items are left unassigned to clusters, while setting pamStage to FALSE allows unclustered items. Finally, *minClusterSize* determines the minimum size of the produced clusters, and thus passing a higher value to this argument prevents the generation of too many clusters with a very small number of items.

This initial set of clusters is to be used as "hooks" to gather as much expression profile diversity from the data as possible. Importantly, even though our parametrization allows isoforms to remain unassigned to clusters (see above), some isoforms may still show low similarity to their cluster's profile. To be able to obtain profiles as consistent as possible for downstream refinement, a cluster quality control step was included in *acorde* to remove isoforms based on a minimum correlation threshold with the rest of the members. For our study, isoforms were moved to the unclustered group if they showed a correlation lower than 0.85 with 3 or more isoforms from their cluster. In this manner, only tightly-correlated groups of isoforms will remain clustered.

*Expanding clusters with unassigned isoforms.* To re-assign unclustered isoforms to clusters with which they show high correlation, *acorde* allows correlation-based cluster expansion. In this process, each cluster profile is summarized into an average representative transcript, hereby referred to as "metatranscript". Meta-transcripts are calculated as the mean of the percentile-based expression of all isoforms in the cluster. As a result, $11 \cdot K$ ($K$ being the number of cell types) mean-summarized percentile expression values are obtained, which can be understood as an approximation to the expression range shown by the isoforms from that cluster in each of the cell types. Next, correlations between metatranscripts and unclustered isoforms are computed and unclustered isoforms assigned if they show percentile correlation values above a specified threshold with at least one cluster. In the present study, we set a correlation threshold of 0.9 with at least one cluster, where the maximally correlated cluster was selected as the best match if there were ties.

*Merging clusters by profile similarity.* Prioritizing the reduction of within-cluster variability may lead to obtaining a large number of small, redundant clusters (Supplementary Fig. 3a). To mitigate this effect while also preserving high correlations between cluster members, *acorde* can be used to merge clusters by profile similarity using the percentile correlations between their metatranscripts. For this study, hierarchical clustering was performed on the metatranscript correlation matrix via the *hclust()* function (*stats* R package), subsequently creating clusters with the *cutree()* function (*stats* R package) and a height cutoff of 0.1. Since this merging process may result in joining clusters with highly uncorrelated profiles (Supplementary Fig. 3b), the cluster expansion process described above was used for the re-assignment of isoforms from clusters that were flagged as inconsistent.

*Recursive assignment of remaining unclustered isoforms.* First, extant clusters were filtered again to maximize similarities between members of the same isoform group and generate reliable profiles for expansion. In this case, isoforms were returned to the unclustered group if they had percentile correlation lower than 0.7 with 10 or more isoforms of their cluster. Next, and following the cluster expansion process described above, percentile correlations between the isoforms to be assigned and cluster metatranscripts were computed. In this case, however, assignment was performed as a recursive process, in which (1) isoforms joined a cluster based on percentile correlation with its metatranscript, (2) metatranscripts were re-calculated for the newly expanded clusters, and (3) assignment was performed again for the remaining unclustered isoforms. The percentile correlation thresholds was sequentially lowered from 0.9 to 0.8 and 0.7. Finally, any isoforms remaining unclustered at this point were assigned to the clusters with which they presented maximum correlation. In doing this, unclustered isoform groups were assigned in order, and highly correlated elements therefore contributed to strengthen within-cluster similarities before assigning more lowly correlated elements.

Finally, expanded clusters were merged again to remove remaining redundancies and generate larger clusters for coDIU detection. The strategy and parameters used were similar to those detailed in the section above.

**Co-expression pattern simulation.** To validate percentile correlations and our clustering strategy, we evaluated their performance on synthetic data, where co-expression relationships between simulated features need to be pre-defined as part of the data simulation process. However, there is, to the best of our knowledge, no currently available strategy to simulate single-cell data including modules of co-expressed features. We therefore designed our own simulation strategy by combining the *SymSim* R package[59] to adequately model single-cell RNA-Seq data, and a dedicated strategy to generate co-expression between *SymSim* simulated features.

First, we set the following parameters to the *SimulateTrueCounts()* function in *SymSim* in order to obtain a count matrix consisting in 1000 cells from 8 cell types and 8000 features, with sufficient feature-level variation between the different cell groups:

*SimulateTrueCounts(ncells_total = 1000, min_popsize = 100, i_minpop = 1, ngenes = 8000, nevf = 10, n_de_evf = 9, evf_type = "discrete", phyla = pbtree(n = 7, type = "discrete"), vary = "s", Sigma = 0.25, gene_effect_prob = 0.5, bimod = 0.4, prop_hge = 0.03, mean_hge = 5).*

Next, we modeled technical effects on these true counts in order to obtain real, observed counts using the *True2ObservedCounts()* function in *SymSim*, with the following parameters:

*True2ObservedCounts(true_counts$counts, meta_cell = true_counts$cell_meta, protocol = "nonUMI", alpha_mean = 0.1, alpha_sd = 0.005, lenslope = 0, gene_len = rep(1000, nrow(true_counts$counts)), depth_mean = 4e6, depth_sd = 1e4).*

To create co-expression patterns, we then re-ranked expression values on a cell type-specific manner to define synthetic features, based on the expression profile of 15 pre-defined co-expression modules.

First, we drafted 15 different co-expression profiles reflecting three levels of expression complexity, that is, showing high expression or expression "peaks" in one, two, or three cell groups, respectively. To generate a count matrix reflecting these expression patterns, we shuffled simulated counts to create new, synthetic features. To achieve this, we first re-ranked features in each cell group by mean expression across cells in the group, breaking feature connectivity between the simulated cell types. Then, the top 1400 features from each cell type were selected, together with the bottom 1400 features. In this manner, we obtained high-expression and low-expression count vectors for each group, which we then combined to create synthetic features following the pre-designed cluster's co-expression pattern. For each cluster, 200 count vectors from top-expression features were assigned to peaking groups, and 200 count vectors form bottom-expression features to cell groups showing low expression. Of note, 1400 features were selected for simulation in order to grant at least 7 different 200-feature groups could be generated for each cell type, where the cell group with the highest-peaking frequency across clusters showed high expression in 6 clusters only.

All in all, we obtained a simulated count matrix containing 1000 cells from 8 cell types and 3000 synthetic features, all of which belong to one of the 15 simulated co-expression modules. Therefore, by breaking feature-level connectivity between cell types, we benefited from feature-specific properties at the cell type level, while re-creating cell type expression coordination patterns that the SymSim strategy was not able to generate. Finally, to ensure the quality of the simulated clusters, we filtered synthetic features if their Pearson correlation with the cluster's median profile was below 0.75 (Supplementary Fig. 4b).

**Benchmarking of isoform correlation metrics for scRNA-seq data.** Traditional correlation metrics have been shown to perform poorly when applied to scRNA-seq data, mainly given the increased noise and stochasticity levels in this data type. Recently, extensive benchmarks including single cell-tailored metrics have shed light on how to best select correlation metrics for single-cell data (see review by Skinnider et al.[42]). We, therefore, compared the performance of percentile correlations to a representative set of correlation metrics used in single-cell co-expression studies, namely classical Pearson and Spearman correlations, single-cell designed zero-inflated Kendall[57] correlation, and proportionality metric rho ($\rho$)[58], in agreement with previous reports showing that proportionality metrics were among the best performing co-expression methods in single-cell data. To measure performance, we computed these five co-expression metrics for all the synthetic features in the previously-simulated dataset, generating five different distance matrices for clustering, and evaluated which metric best recapitulated the simulated co-expression modules when used in our clustering pipeline. Pearson and Spearman correlations were computed using the *cor()* function in the R-base *stats* package. Zero-inflated Kendall correlation and rho ($\rho$) were computed using the *dismay()* function in the *dismay* R package[42].

To make our benchmarking comparable, we adapted our clustering pipeline to remove all non-automated steps and always generate a fixed number of clusters. First, hierarchical clustering was performed on each correlation matrix using *dynamicTreeCut()*[56] and the following non-default parameters to maximize granularity: *deepSplit = 4, pamStage = FALSE, minClusterSize = 10*. Of note, we skipped the quality filtering step based on intra-cluster correlations (see isoform clustering section above) to avoid bias against metrics that tend to yield low values when applied to single-cell data. Since we intended to evaluate the number of

features remaining unclustered using each metric, we additionally suppressed the unclustered isoform assignment step (see isoform clustering section above). Finally, the merge process was automated by using the traditional hierarchical clustering algorithm (implemented in the *hclust()* function in the R stats package) to group clusters based on the inferred metatranscripts that summarize the cluster's expression profile (see isoform clustering section above). Finally, we set the number of clusters to 15, i.e., the number of simulated co-expression modules.

In addition to the number of unclustered isoforms, we used the levels of internal correlation in the empirical clusters, i.e., those obtained by de novo clustering of simulated synthetic features, to evaluate the clustering. We did this by jointly considering all pairwise metric values for features within a cluster and measuring the percentage of metrics that are above a threshold value of 0.8. To assess how well empirical clusters recapitulated the co-expression simulation, we paired empirical with simulated clusters using the correlations between their mean cluster profiles. Simulated clusters were therefore paired with the empirical clustering showing maximum profile correlation. We next compared synthetic feature IDs assigned to the obtained and empirical clusters in each pair using the Jaccard Index (JI).

### Differential isoform usage and co-Differential isoform usage across multiple groups

*Defining DIU across multiple groups.* Grouping isoforms into different clusters allows the detection of a number of expression patterns across the multiple cell types included in single-cell data. As previously described, we filtered DE isoforms to ensure that all transcripts had at least one other counterpart from the same gene that was also significantly DE. Intuitively, in order for Differential Isoform Usage (DIU) to occur, a gene must first have at least two DE isoforms. However, we only considered a gene to be positive for DIU if (at least) two isoforms were DE and were assigned to different clusters, indicating that two of the gene's isoforms show different expression patterns across groups (see main text, Fig. 4a). Ultimately, this can be interpreted as an indicator that isoform expression regulation is cell type-dependent in that gene.

*Detecting co-splicing patterns across isoform clusters: co-Differential isoform usage.* We define coordinated splicing patterns as a situation where post-transcriptional regulation, defined by isoform expression, can be detected independently of transcriptional regulation, i.e., gene-level expression. To detect splicing coordination, we defined coDIU as a pattern where a group of genes shows co-expression of their isoforms, but no co-expression can be detected when only gene expression is considered (see main text, Fig. 4b). In the context of our pipeline, a set of potentially coDIU genes will have at least two of their isoforms assigned to the same clusters, therefore showing detectable isoform-level co-expression, and suggesting coordinated splicing regulation in that group of genes. However, clustering allows expression pattern variability among members, and therefore some isoforms might be assigned to clusters that do not faithfully represent their expression profile, leading to the detection of false-positive coDIU genes.

To identify groups of genes candidates for coDIU, we applied negative-binomial generalized linear regression models. Let $G$ be a group of genes, each of them with $I_g$ isoforms, where $g = 1,…,|G|$. At least one of the isoforms of each gene $g$ in $G$ must belong to the same cluster $c$, where $c \in \{1,…,C\}$ and $C$ is the total number of clusters. Let $\mathbf{z}$ be the expression vector obtained after concatenating the expression vectors $\mathbf{y_i}$ of each isoform $i$ of every gene $g = 1,…,|G|$. For the sake of simplicity, let us assume that $|G| = 2$, $I_g = 2$ $\forall g$, and consequently $C = 2$. In this case, vector $\mathbf{z}$ will contain $4N$ elements, where $N$ is the total number of cells in the data ($N = 1591$ in our data) and will be the response variable in our regression model. We need to assess if $\mathbf{z}$ values follow the trend depicted in Fig. 4b, that is, the average profile across cell types of the two isoforms in cluster 1 must be significantly different to the average profile of the two isoforms in cluster 2. In addition, the average profile of the two isoforms of gene 1 must not be different to the average profile of the two isoforms of gene 2. To identify groups of genes with these characteristics we proposed to fit the regression model in Eq. 4 and select the group of tested genes as coDIU candidates when having a significant interaction between cluster and cell type effects, and a non-significant interaction between gene and cell type effects.

$$h(z) = \beta_0 + \beta_1 G_2 + \beta_2 C_2 + \sum_{k=2}^{K} \gamma_k T_k + \beta_3 G_2 C_2 \\ + \sum_{k=2}^{K} \delta_k T_k G_2 + \sum_{k=2}^{K} \tau_k T_k C_2 + \varepsilon \quad (4)$$

Where $G_2$ and $C_2$ are dummy variables indicating whether the expression value corresponds to gene or cluster 2 (value 1) or 1 (value 0), respectively, $T_k$ is a dummy variable which takes value 1 when the corresponding cell is assigned to cell type $k$ ($k = 1,…,K$) and 0 otherwise, $\beta_k$, $\gamma_k$, $\delta_k$ and $\tau_k$ are the regression coefficients, $\varepsilon$ represents the error term, and $h()$ is the link function of the GLM (natural logarithm in this case).

We fitted the GLM model with the glm() function in the R-base package, and the *negative.binomial()* function in the *MASS* R package[92], with $\theta = 10$. To test the significance level of the *cluster\*cell type* and *gene\*cell type* interactions, we calculated type-II analysis-of-variance (ANOVA) tables for the model using a likelihood-ratio $X^2$ test, implemented in the *Anova()* function from the *car* R package[93], since we had an unbalanced design. *P*-values for each of the interactions were separately adjusted using the Benjamini & Hochberg correction. Gene pairs were considered positive for coDIU if FDR adjusted *p*-value < 0.05 for the

*cluster\*cell type* interaction and FDR adjusted *p*-value > 0.05 for the *gene\*cell type* interaction. In other words, we required expression variance across cell types to be a function of the expression profile captured by the clustering, while imposing the additional limitation that aggregating expression by gene must make this effect undetectable. Given that all genes with clustered isoforms will form pairs with all potentially coDIU counterparts and be repeatedly tested, we considered genes to be positive for coDIU if they met the significance criteria in at least one of these pairwise tests.

### Functional analyses.

The analyses in this manuscript are based on a long read-defined transcriptome which, after careful quality control and curation of the isoform models, was further annotated using IsoAnnotLite (https://isoannot.tappas.org/isoannot-lite/) to include positionally-defined functional features in the annotation (see Supplementary Note). *Functional features* are grouped in *functional categories* depending on the database from which the information was retrieved and on the biological functions performed by the features (comprehensive list in Supplementary Note). In this manner, we gathered sufficient information to couple our co-expression analyses with a biological readout. The specific analysis strategies used to this end are detailed below.

*Functional enrichment analysis.* In order to understand the functional properties of AS-regulated and co-regulated genes, we set out to characterize DIU and coDIU genes using different functional enrichment analysis approaches. In this manner, we intended to gain insight on functional features and categories showing significant overrepresentation in each of these two gene lists, in comparison to different backgrounds, i.e., lists of genes to compare to in order to detect enrichment.

In the case of DIU genes, we calculated enrichment relative to genes with multiple DE isoforms in order to discriminate the functional properties of genes regulated by AS, as opposed to those lacking differential usage of their isoforms. We considered all annotated functional categories and features, and applied tappAS Functional Enrichment Analysis (FEA), which relies on the *GOSeq* R package[94]. Briefly, the method performs an over-representation Fisher's Exact test for each functional feature, considering the number of genes annotated with the feature in the tests and background lists. tappAS next corrects for multiple testing within each functional category by the Benjamini-Hochberg method, allowing multiple functional databases to be included or excluded from the analysis without influencing the number of significant features after *p*-value adjustment. Significant enrichment for the different tests was defined using a threshold of FDR < 0.05.

For coDIU genes, we designed a different strategy in order to improve the statistical power of our functional enrichment analysis, aiming to compare functional properties between splicing regulation (DIU) and co-regulation (coDIU). As stated above, DIU regulation is best measured by using genes with DE isoforms as background. Intuitively, coDIU-regulated genes should then be characterized by comparing them to DIU genes. To accommodate these two test/background lists in a functional enrichment analysis without ignoring the overlap between the coDIU and DIU gene groups, we computed enrichment using a partially overlapping samples z-test via the *Prop.test()* function in the *Partiallyoverlapping* R package[66]. Specifically, we compared the proportion of coDIU genes containing each of the functional features (relative to DIU genes) with the proportion of DIU genes containing that same annotation (with respect to genes with DE isoforms). In other words, we tested whether the proportion of coDIU vs DIU genes including a given functional feature was significantly higher than that shown in the comparison between DIU and DE genes. We performed the analysis for features with more than 15 annotated genes, and subsequently corrected for multiple testing within functional categories using the Benjamini-Hochberg method. For GO terms, ontologies with more than 150 annotated genes were also removed to eliminate excessively broad—and potentially less meaningful-functions. Functional features were considered to be present in a significantly higher proportion in coDIU genes when FDR < 0.05.

Annotations used in all functional analyses included Gene Ontology (GO) terms. The hierarchical structure of the Gene Ontology database can often result in multiple significantly enriched terms that refer to the same, or very similar, functions, components, and processes. To enhance visualization and result interpretation of coDIU functional enrichment results, we used Revigo[67] to perform a semantic similarity analysis of all significant GO terms obtained in the partially overlapping samples test. We applied a dispensability (i.e., measure of semantic similarity) threshold of 0.5 to assign GO terms to a cluster, and then selected a representative of each term cluster to include in visualization.

*Functional diversity analysis.* To obtain insight into the functional changes generated as a consequence of DIU and coDIU, we again used the tappAS tool for the functional analysis of AS[61]. In particular, we first applied tappAS' Functional Diversity Analysis (FDA) module (see main text Fig. 5c). Briefly, FDA performs a within-gene comparison of all the isoforms included in the analysis, aiming to detect whether they present variation in the inclusion of a functional feature. In FDA, variation can be positional, i.e., one or more of the gene's isoforms present a change in the genomic coordinates defining the feature, or be defined by presence/absence, i.e., at least one of the isoforms lacks a feature that is present in the rest. As a result, FDA provides analyzed genes with a label for each of the feature categories included in the transcriptome's functional annotation file, flagging them as *varying*

if at least one of the isoforms presents the variation in a feature from that category, or *not varying* if no changes are detected. For more details on FDA, see the Methods section in de la Fuente et al.[61].

We run both positional and presence/absence FDA for three gene sets: (1) genes with multiple DE isoforms, (2) DIU genes, and (3) coDIU genes. Next, for each of these gene sets, we computed the proportion of *varying* genes detected for each functional category. *Varying* proportions were calculated relative to the total number of genes including annotations from the category, instead of considering all genes in the set. In this manner, we avoided underestimating variation rates for categories that were less represented in the functional annotation file. In order to check whether any of these gene sets presented a significantly higher mean proportion of *varying* genes across categories, we performed a paired *t*-test for each combination of gene set pairs: DIU vs multiple DE, coDIU vs multiple DE, and coDIU vs DIU. In this analysis, we considered functional categories to be the individuals under evaluation, while the proportion of *varying* genes calculated for each category in the two tested sets constituted the paired observations. As a result, we obtained three *p*-values per FDA analysis type, i.e., presence/absence and positional variation.

To better understand the functional readout that can be obtained using the *acorde* pipeline, we analyzed a subset of the coDIU gene network, namely three clusters showing related isoform co-expression patterns: neuron-specific expression (cluster 1), oligodendrocyte-specific expression (cluster 14) and expression in both neural and oligodendrocyte cell types (cluster 4). To characterize functional variation among the clusters, we used positional/presence FDA (see above) and ID-level FDA. ID-level FDA is also included in tappAS[61] and provides a within-feature summary of FDA results. In other words, ID-level FDA ultimately reports the number of *varying* and *not varying* genes detected for each feature ID included in a given functional category. In this case, *varying* status obeys a similar criterion to the one described above, i.e., genes in which at least one isoform shows differential inclusion/exclusion of the feature. Since each functional category may include several features, ID-level FDA provides a complementary view to that of FDA, allowing users to inspect which particular features are more frequently changing as a result of the category-level functional variation reported in FDA. For more details on ID-level FDA, see the Methods section in de la Fuente et al.[61].

**Analysis of GABA-ergic neuron subtypes**. To illustrate the applicability of the *acorde* approach, we retrieved additional single-cell RNA-Seq data from a second study by Tasic et al.[44]. Single-cell libraries belonging to GABA-ergic neuron cell subtypes from primary visual cortex were retrieved from SRA accession SRP150473 (https://www.ncbi.nlm.nih.gov/sra/?term=SRP150473) using the cell identity information in GEO accession GSE115746 (https://www.ncbi.nlm.nih.gov/geo/query/acc.cgi?acc=GSE115746), i.e., 6147 cells and 7 cell subtypes in total.

Cell-level isoform expression estimates were obtained with Kallisto[95] using our long read-generated neural transcriptome (see Supplementary Note). Cell clusters and the cell subtype labels assigned to them by authors in the original study were retrieved and used in the analysis. After quality filtering of cells (1.25e6 < total counts < 2.75e6) and lowly expressed isoforms (counts > 0 in at least 25% of one cell type), we additionally removed isoforms that did not accumulate more than 10% of their gene's expression in at least one cell type. Cells belonging to cell types *Meis2* ($n = 43$) and *Serpinf1* ($n = 22$) were discarded to balance cell type abundances, as the remaining cell types had ~1000 cells each. Differentially Expressed (DE) isoforms were next computed by combining ZinBWAvE weights[48] (see prior DE section) and DE testing using edgeR[50]. Isoforms with FDR < 0.05 and fold-change > 1.5 between at least one pair of cell types were selected for downstream analysis. After computing percentile correlations with $p = 10$, DE isoforms were clustered using dynamic hierarchical clustering with the following non-default parameters for the *cutreeHybrid()* function in the *dynamicTreeCut* R package:[56] *deepSplit = 4, pamStage = FALSE, minClusterSize = 2, cutHeight = 0.1.* Parameters were fine-tuned to generate a high number of clusters with as accurate a profile as possible. Unclustered isoforms (34 isoforms in total) were assigned to clusters using the percentile correlation with cluster metatranscripts and successively decreasing thresholds (see section on cluster expansion above). Expanded clusters were then merged by dynamic clustering of their metatranscripts (see Methods section on merging clusters) using the following non-default parameters for the *cutreeHybrid()* function: *pamStage = FALSE, cutHeight = 0.3.* Detection of DIU and coDIU genes was performed as described in the relevant Methods sections.

**Reporting summary**. Further information on research design is available in the Nature Research Reporting Summary linked to this article.

## Data availability

Single-cell, short-read RNA-Seq data from mouse primary visual cortex used in the analysis of neural broad types, generated by Tasic et al. 2016[43], was downloaded from SRA accession SRP061902. Single-cell, short-read RNA-Seq data from mouse primary visual cortex used in the analysis of GABA neuron subtypes, generated by Tasic et al. 2018[44], was downloaded from SRA accession SRP150473, after selecting accessions corresponding to GABA neurons and primary visual cortex tissue, described by authors in the study metadata available at GEO accession GSE115746. The isoform-level expression matrices

obtained using these data and the long read-defined transcriptome are available as data objects within the *acorde* R package (https://github.com/ConesaLab/acorde/data). Mouse reference genome and transcriptome used for long-read processing were downloaded from the RefSeq[96] database (global release 96, annotation release 108, September 2019, https://ftp.ncbi.nlm.nih.gov/genomes/all/annotation_releases/10090/108), from genome version GRCm38.p6 and assembly accession GCF_00001635.26. Long-read datasets form mouse hippocampus and cortex, generated by Wyman et al.[45], were downloaded from ENCODE accessions ENCSR214HSG and ENCSR340GWV, respectively. Long read-defined transcriptome files (generated using above-cited long-read data and reference files, details in Supplementary Note) have been made available at the tappAS repository of annotation files. These include the GTF file used for quantification (https://app.tappas.org/resources/downloads/gtfs) and the GFF3 file obtained after transferring functional features using isoAnnotLite (http://app.tappas.org/resources/downloads/gffs); indicated as Mus_Musculus_GRCm38.p6_PacBioENCODE_RefSeq108.

## Code availability

The code used to perform the analyses in this manuscript has been implemented in the *acorde* R package, available at https://github.com/ConesaLab/acorde. Specifically, all analyses have been run using *acorde* v0.1.0[97].

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

## Acknowledgements

This work has been funded by NIH grant R21HG011280 (A.C.) and by the Spanish Ministry of Science grants BIO2015–1658-R (A.C., S.T.), BES-2016-076994 (A.A.L.) and PID2020-119537RB-100 (A.C., S.T.). Funding for open access charge provided by the Universitat Politècnica de València and the Spanish Ministry of Science grant PID2020-119537RB-100.

## Author contributions

A.A.L. developed and evaluated computational methods, implemented and documented R package, obtained long read transcriptome, analyzed single-cell data, interpreted results, and wrote the manuscript. P.S. implemented IsoAnnotLite, refined and performed functional annotation strategy. S.T. conducted statistical method development, refined the manuscript, and supervised the study. A.C. conceived the study, supervised all analysis approaches and methods developed, and refined the manuscript.

## Competing interests

The authors declare no competing interests.
