## [Peer Review File · Nature Communications]

Reviewers' Comments:

Reviewer #1:

Remarks to the Author:

The authors presented *acorde*, which performs multiple steps to obtain isoform co-expression modules and genes with DIU (differential isoform usages) or coDIU. The main technical contributions are: 1) mapping to a reference created with bulk long read data for isoform expression quantification; 2) using "percentile correlation" measure to calculate correlation between isoforms; 3) building a graph between isoform clusters to identify DIU and coDIU genes. Functional analysis of these DIU and coDIU genes is performed. The resulting GO terms and some example results from the FDA (functional diversity analysis) serve as evidence to show that the pipeline leads to meaningful biological hypotheses/discussions. Studying patterns of isoform usage in single cells is a relatively under-studied area and appropriate computational tools are needed. While this pipeline contains interesting novel ideas, it needs to be improved from multiple aspects.

Overall, it's not well supported that each of the 3 steps (especially the first 2) above plays a unique role to lead to the final results. Either benchmarking each step better (see points 1 and 4 below), or showing that if any of steps 1 and 2 is replaced with an existing alternative, then the pipeline won't yield some of the interesting results that *acorde* finds would make the results stronger. Please see the specific points below:

1. For the first step of *acorde*, which is to quantify isoform expression, the authors proposed to first create a reference using bulk long reads data, which is conceptually a good idea, but no data is provided to show that this is superior to using a standard reference.
2. Page 7: the authors downsampled cells in the cell types which originally have more cells, and as a result, the number of cells in each cell type becomes very small (~30). This practice has introduced extra randomness in the DE results due to downsampling, as discussed on pages 7 and 8. However, it's not clear what's the benefit of performing the downsampling. The DE methods do not require that cell types have similar size. The authors wrote the motivation as "In order to facilitate downstream analysis" -- could the authors please elaborate how the downsampling facilitates downstream analysis?
3. Related to the point above: the authors checked the difference of resulting DE genes by DESeq2 and edgeR respectively from multiple runs of downsampling. However, as the final set of DE genes/isoforms is the union of the DESeq2 and edgeR results, it would make more sense to investigate the difference of the union of the results from the two DE methods. That is, for each downsampling run, obtain the union of the DESeq2 and edgeR results, and calculate consistency measures like Jaccard Index based on the union sets. The mean Jaccard Index of around 0.74 is a bit worrisome (Page 8, line 5) and that means there can be a large number of isoforms which vary between different downsampling runs. If downsampling is not avoidable, the authors may want to consider taking a consensus set (eg. using majority voting) of DE isoforms from multiple downsampling runs instead of using the results from just one run.
4. The percentile correlation is a major technical contribution of this manuscript. It is also an important step which can largely change the results from downstream steps of using it to find isoform clusters, the cell type-level co-expression network, the CIU and coDIU genes. I think this measure needs to be further investigated. First, when comparing the percentile correlation with other correlation measurements (Pearson, Spearman, etc) on simulated data, the results (Supp. Fig. 5E) show that percentile correlation gives overall higher correlation values for within-cluster isoform pairs. So how do the measures work with isoform pairs which are not supposed to be in the same cluster by simulation? Whether a metric can distinguish more correlated pairs from less correlated pairs is more important how large the value it returns is. If percentile correlation can distinguish isoform pairs from the same clusters from those from different clusters better than other metrics, then it's more convincing. Since various correlation measures are extensively used, I would check the statistical significance (p-value) of each calculated correlation value from each measure, and keep only the ones with low p-values (eg. <0.05) for further comparison.
5. Also in the part of testing percentile correlation with simulated data, some permutation and re-

selection is done based on the simulated data from an existing simulator to create co-expressed isoform clusters. It's not clear whether the original cell type structure is still preserved after the permutation and re-selection. It's important to show this in order to proceed with the same cell type information.

6. This comment is about how automated and complete the package is. The authors mentioned that this is an "end-to-end" pipeline multiple times -- not quite sure what "end-to-end" means here, but the pipeline does involve manual steps seen from both the manuscript and the vignette. A main one is when merging the isoform clusters, the "visibly redundant clusters" (from vignette) were merged using a manually entered list. (It is indeed acknowledged on page 5, lines 8-9 that the clustering approach is "semi-automated".) Also, although in the manuscript and Figure 1 it is described that the *acorde* workflow includes the functional annotation and functional analysis parts (which are the key modules to test whether the whole pipeline leads to biologically meaningful results), the vignette stops at the step of returning DIU and coDIU genes, and it doesn't seem that the github package integrates the functions for functional analysis.

Minor points:

1. Page 5, line 10: the authors used "re-defined" which indicates there exists a previously defined version of the terms DIU and coDIU but there are no references here.

2. Page 5, line 19: by "cell type-specific co-expression of isoforms ...", do the authors mean "cell type-level co-expression of isoforms ..." or something else? "cell type-specific co-expression of isoforms" would mean isoforms that co-expression in certain cell types but not in other cell types to me, which is certainly interesting, but isn't seem to be performed in this work.

3. Page 18, line 18: the authors use "oligodendrocyte-specific" and "neuron-specific" here (or oligodendrocytes vs neurons) and at other locations of the manuscript. It would be helpful for the readers if the authors can specify which of the 7 cell types defined on pages 6 and 7 each of these two categories (oligodendrocytes and neurons) correspond to.

Reviewer #2:

Remarks to the Author:

In this manuscript, Arzalluz-Luque et al proposed a computational pipeline, *acorde*, to identify isoform co-usage network from single-cell transcriptomics data with a newly proposed percentile correlation method and to perform functional analysis of the clusters of genes with similar co-usage patterns. The splicing isoform analysis in single cells is an interesting topic; this paper has a focus on the functional analysis of co-used isoforms instead of the splicing's impact on cell heterogeneity.

The *acorde* R package could be a useful pipeline, but the novelty on the methodology in each step is not high. Instead, this manuscript is more a demonstration that the proposed analysis strategy of isoform co-usage can reveal biological functions, in a context of cell types, which can also be highly valuable. However, this manuscript is only based on one dataset, so it is hard to evaluate the generalizability of the proposed pipeline, particularly given there are multiple settings and thresholds to tune in each step of the pipeline. Therefore, it would be more insightful to apply the pipeline to more datasets and demonstrate the usefulness of these analyses, even without long-reads data to pre-filter the isoform annotations.

The manuscript is well written and very detailed. Here, I have a few comments on the technical part:

1. The proposed percentile correlation is an appealing metric to measure the correlation between transcripts across cell types. From my understanding, it is an extension of median value of cells in a cell type. Instead of only using median value (a single percentile), this method uses 10 percentile values to represent the isoform expression in each cell type for calculating a Pearson Correlation Coefficient (PCC) of two isoforms across all cell types. Therefore, a more relevant

comparison is to the median value of each cell type, so it can show how much the percentile correlation improves from median correlation, and it can imply their relation. From the Supp Fig. 3C, it seems 4-percentile is comparable to 10-percentile, so single-percentile (median) may be not too far.

2. Since the sample size ($n_{\text{cell_types}} * n_{\text{percentiles}}$) is small, the $PCC > 0.8$ may not necessarily mean a statistical significance. A permutation test would be useful to evaluate the significance, for example by shuffling the cell types of one of the two isoform, e.g., for a thousand times. I would suggest keep the percentiles of each cell type when shuffling them.

3. Related to point 2, in the simulated data (Supp. Fig. 5E), it only reports the proportion of highly correlated isoform pairs but didn't mention false positives. It could be possible that the percentile correlation is a method with high sensitivity, but also has high false positive rate.

4. In p.7, the number of detected DE isoforms are 10,100 out of 13,832, which means 73% transcripts are differentially expressed in at least one cell type comparing to all others. This proportion seems very high. It is important to evaluate how the initial selection of DE isoform impact on the downstream analysis. For example, one can add fold change as additional cutoff besides FDR as often used in DEseq or edgeR and use the overlapped DE isoform instead of the union between the two methods.

5. The clustering of the isoform based on the correlation matrix is of high uncertainty because of both the noise in the correlation matrix and the selection of the number of clusters. This may be the reason why the authors introduce a semi-automated clustering, to adjust the clustering manually. Therefore, it is important to evaluate how the settings in this clustering step will impact on the downstream analysis. For example, based on currently setting (p.14), in 80% of genes have differential isoform usage, which seems too high. So whether a smaller number of isoform clusters may give more stringent DIU.

6. Related to Fig. 5D where the number of coDIU has a linear relation to the number of pairs. It will be useful to report the significance on the n_{coDIU} comparing to random by a permutation test, where two group of isoforms are randomly generated to a matched sizes of the two isoform clusters, and n_{coDIU} under the null can be calculated. By repeating this procedure many times, an empirical p value can be obtained. Both the permutation p values and the distribution of the n_{coDIU} under the null can be informative to present.

Reviewer #3:
None

Reviewer #1 (Remarks to the Author)

The authors presented *acorde*, which performs multiple steps to obtain isoform co-expression modules and genes with DIU (differential isoform usages) or coDIU. The main technical contributions are: 1) mapping to a reference created with bulk long read data for isoform expression quantification; 2) using “percentile correlation” measure to calculate correlation between isoforms; 3) building a graph between isoform clusters to identify DIU and coDIU genes. Functional analysis of these DIU and coDIU genes is performed. The resulting GO terms and some example results from the FDA (functional diversity analysis) serve as evidence to show that the pipeline leads to meaningful biological hypotheses/discussions. Studying patterns of isoform usage in single cells is a relatively under-studied area and appropriate computational tools are needed. While this pipeline contains interesting novel ideas, it needs to be improved from multiple aspects.

Overall, it's not well supported that each of the 3 steps (especially the first 2) above plays a unique role to lead to the final results. Either benchmarking each step better (see points 1 and 4 below), or showing that if any of steps 1 and 2 is replaced with an existing alternative, then the pipeline won't yield some of the interesting results that *acorde* finds would make the results stronger. Please see the specific points below:

1. For the first step of *acorde*, which is to quantify isoform expression, the authors proposed to first create a reference using bulk long reads data, which is conceptually a good idea, but no data is provided to show that this is superior to using a standard reference.

We appreciate the reviewer's concern. We are aware that transcriptomics analyses are generally based on standard reference transcriptomes and, while our isoform co-expression analysis is, in principle, independent of the reference transcriptome used for quantification, we chose to use a long-reads-based reference as previous studies have shown an increased quantification accuracy in this case.

In our introduction, we cite Sonesson et al. (<https://doi.org/10.1186/s13059-015-0862-3>), who demonstrated both that incomplete reference annotations have a detrimental effect when computing differential transcript usage and that filtering non-expressed isoforms from the reference prior to quantification improved the False Discovery Rate (FDR) when calling differentially spliced genes. In Tardáguila et al. (<https://doi.org/10.1101/gr.222976.117>), our group described and validated (via qPCR) several cases where RefSeq-based quantification generated expression estimates that were inconsistent with the read coverage pattern, which in turn fitted other novel, PacBio-detected transcripts originally absent from the reference. Extending this result to the whole transcriptome, this study also showed that, for ~10% of analyzed genes, the most highly expressed isoform was a novel transcript detected by PacBio. These studies therefore demonstrate the benefits of building an *ad hoc* transcriptome for the analyzed tissue or samples for both quantification accuracy and the detection of relevant alternative isoforms that are not included in the reference transcriptome.

To assess whether these observations held in our study, we quantified the RefSeq reference transcriptome using single-cell short-reads from Tasic et al. (2016) and compared the results to our long read-defined transcripts (see Figure 1 below).

We found that 8734 transcripts were commonly identified by both approaches, while 8313 and 12203 were found exclusively by the long-read and Refseq references, respectively. Out of the long-reads exclusive transcripts, 4751 were novel and Incomplete Splice Match (ISM) transcripts, which present novelty at the level of junctions and TSS/TTS in comparison to RefSeq, respectively. Furthermore, these had cell type median expression levels comparable to those shown by Full Splice Matches (FSM), which include transcripts present in both transcriptomes as well as long read-unique FSM. In addition, the expression of these novel isoforms was often higher than that of transcripts uniquely detected in the RefSeq quantification, while RefSeq unique transcripts had higher zero expression frequencies than both FSM and novel/ISM transcripts.

Figure 1: Distribution of median cell type expression (logcounts). Violin plot shows density distributions. Boxplot indicates quantiles and median of cell type-level expression. Zero values have been excluded for optimal visualization (total no. of excluded observations for each category are shown in the upper right-side plot). Results are stratified by transcript type, namely FSM (including those uniquely detected by long reads and in common with RefSeq), novel/ISM (i.e. transcripts representing the novelty introduced by long reads) and RefSeq unique (i.e. isoforms from RefSeq for which no associated FSM is found using long reads).

We would like to note that all long read transcripts have been subject to ML-based filtering with SQANTI and to further validation of novel TSS/TTS using orthogonal data and annotation information (see Supplementary Note 1). In contrast, RefSeq unique isoforms may be detected due to the absence of other, tissue and system-specific isoforms, as discussed by Tardáguila et al. (<https://doi.org/10.1101/gr.222976.117>). We hypothesize that the higher density of cell type zero median values in this subgroup of isoforms may be evidence of this phenomenon.

Finally, we would like to illustrate the relevance of long read-defined isoforms in our study. When analyzing the SQANTI3 categories represented in isoforms participating in coDIU relationships for the Tasic dataset (Tasic et al. 2016, see references), we found that ~30% of isoforms belonged to novel SQANTI categories. This means that one third of co-expressed isoforms in our network were novel transcripts detected using PacBio, highlighting the discovery potential of combining long read-based isoform models with the co-expression analyses implemented in *acorde*.

In summary, although the *acorde* isoform co-expression approach is in principle agnostic to the reference used for transcript quantification, our method benefits from using a sample-specific long reads transcriptome, due to the improved detection of the expressed isoforms. We believe that the discussion and citation of Sonesson et al. and Tardáguila et al., already included in our introduction (page 6, lines 5-10) provides sufficient reference to previous work to support the utilization of a long read-based transcriptome in the present analysis.

2. Page 7: the authors downsampled cells in the cell types which originally have more cells, and as a result, the number of cells in each cell type becomes very small (~30). This practice has introduced extra randomness in the DE results due to downsampling, as discussed on pages 7 and 8. However, it's not clear what's the benefit of performing the downsampling. The DE methods do not require that cell types have similar size. The authors wrote the motivation as "In order to facilitate downstream analysis" -- could the authors please elaborate how the downsampling facilitates downstream analysis?

We agree with the concerns raised by the reviewer in this comment. We would like to point out that, in the paper, Differential Expression analysis is intended as a pre-filter to ensure the robustness of clustering and coDIU results. Therefore, we ultimately decided to minimize the amount of information regarding downsampling justification, mainly for the sake of manuscript length, but also to be able to focus on the analyses that are, in our view, more substantial to the manuscript. Nevertheless, we are happy to elaborate more on the subject in this response to reviewers.

We encountered two different problems when performing DE analyses using the full Tasic dataset. In the case of DESeq2, the process was too computationally costly and failed to finish running on an 8 core, 64GB machine. Regarding edgeR, the analysis yielded 11455 transcripts as significantly DE (AdjPValue < 0.05), which represent 85% of the transcripts that remained after quality control. We attributed this to simple size differences between glial cell types and abundantly represented neural types, which present internal transcriptomic differences that facilitate the detection of DE transcripts. Downsampling neural cell types to 45 cells allowed us to balance sample size across cell types, while simultaneously avoiding the detection of too many DE transcripts for downstream analysis. On a side note, we would like to highlight that, among the 5711 transcripts detected as DE by edgeR for our downsampled data, 98% were part of the larger set of DE features obtained in the full-dataset edgeR run, suggesting that downsampling was unlikely to yield false positive DE transcripts.

3. Related to the point above: the authors checked the difference of resulting DE genes by DESeq2 and edgeR respectively from multiple runs of downsampling. However, as the final set of DE genes/isoforms is the union of the DESeq2 and edgeR results, it would make more sense to investigate the difference of the union of the results from the two DE methods. That is, for each downsampling run, obtain the union of the DESeq2 and edgeR results, and calculate consistency measures like Jaccard Index based on the union sets. The mean Jaccard Index of around 0.74 is a bit worrisome (Page 8, line 5) and that means there can be a large number of isoforms which vary between different downsampling runs. If downsampling is not avoidable, the authors may want to consider taking a consensus set (eg. using majority voting) of DE isoforms from multiple downsampling runs instead of using the results from just one run.

In spite of our response above, we agree with the reviewer's concerns regarding the extra randomness introduced by selecting DE transcripts based on just one downsampling run. To alleviate this effect, we followed the reviewer's suggestion to instead take a consensus set of DE isoforms. Briefly, we computed the union between results of both DE methods across 50 different downsampling runs, and selected transcripts that were detected as significantly DE ($\text{AdjPValue} < 0.05$) at least 50% of times. Reassuringly, the mean Jaccard Index across all possible combinations of these 50 sets of DE transcripts (union of edgeR and DESeq2 results) was 0.82, with a standard deviation of 0.009. As a result, we retained 9393 transcripts for downstream analysis, and computed percentile correlations between them using the full dataset (i.e. downsampled data was not used from this point on). The details of the new DE analysis are fully described in the Methods section of the re-submitted version of our manuscript, and the results section regarding DE analysis has been amended in page 7.

Of note, changing the set of DE transcripts and abandoning the downsampling strategy required re-running the entire *acorde* pipeline, which has led to modifications in results, discussion, figures and methods sections. Even though we have done our best to highlight all of these changes in the new version of the manuscript, we encourage the reviewer to take this into account when evaluating the work again.

4. The percentile correlation is a major technical contribution of this manuscript. It is also an important step which can largely change the results from downstream steps of using it to find isoform clusters, the cell type-level co-expression network, the CIU and coDIU genes. I think this measure needs to be further investigated. First, when comparing the percentile correlation with other correlation measurements (Pearson, Spearman, etc) on simulated data, the results (Supp. Fig. 5E) show that percentile correlation gives overall higher correlation values for within-cluster isoform pairs. So how do the measures work with isoform pairs which are not supposed to be in the same cluster by simulation?

We thank the reviewer for pointing this out. We agree that showing correlation values for inter-cluster isoform pairs would be a valuable addition to the paper to better understand percentile correlation. Therefore, we have calculated the intra-cluster and inter-cluster correlation for all metrics and for each of the 15 clusters generated by simulation, as suggested. Given that it would not be as useful to include 15 similar figures in the paper, we have created a summarized version in which all within and inter-cluster correlations are aggregated across clusters, showing only one plot per metric. This figure is included in the

manuscript as a Supplementary Figure 5F, and discussed in the main text in page 11, lines 20-23. We believe this provides a sufficient account of the behaviour of each of the metrics when isoforms pairs from the same and different simulated clusters are compared. In these analyses, we observe overall good separation between pairs that are supposed to cluster together by simulation and those which are not, especially for rho and percentile correlations. Remarkably, we find that our metric gives the overall best separation between correlation density distributions, particularly when all clusters are jointly considered (Supplementary Figure 5F).

Whether a metric can distinguish more correlated pairs from less correlated pairs is more important how large the value it returns is. If percentile correlation can distinguish isoform pairs from the same clusters from those from different clusters better than other metrics, then it's more convincing. Since various correlation measures are extensively used, I would check the statistical significance (p-value) of each calculated correlation value from each measure, and keep only the ones with low p-values (eg. <0.05) for further comparison.

Although in the case of our analyses correlation is used as a distance metric for clustering, which may pose a challenge when it comes to discarding values, we fully understand the reviewer's concerns regarding potential false-positives among the computed correlations. To evaluate whether the presence of high, spurious correlations, could be a problem in our data, we have computed the significance of the percentile correlations. In particular, we have applied the test implemented in the `cor.test()` function (R stats package) for Pearson correlation to percentile-summarized isoform expression. Our results show that, while all high and medium correlations observed had significant AdjPValues, there was a majority of low correlations that also returned significant AdjPValues (see Figure 2 below). Filtering by correlation significance would therefore be ineffective to discriminate highly correlated isoform pairs.

Figure 2: Significance of percentile + Pearson correlation values. x-axis shows correlation value intervals. y-axis represents the % of computed values in each interval. Color strength indicates whether correlations are above or below a significance threshold of $AdjPvalue = 0.05$.

5. Also in the part of testing percentile correlation with simulated data, some permutation and re-selection is done based on the simulated data from an existing simulator to create co-expressed isoform clusters. It's not clear whether the original cell type structure is still preserved after the permutation and re-selection. It's important to show this in order to proceed with the same cell type information.

As suggested by the reviewer, we have checked that the cell type structure generated by SymSim is not broken or altered when simulating co-expressed isoform clusters. Results can be observed in the form of tSNE plots before and after co-expression simulation, which are now included in the manuscript as Supplementary Figure 5A and discussed in the results section in page 11, line 5. Of note, given that the simulated transcripts possess high variability across cell types to generate distinguishable co-expression patterns, separation between the simulated cell types in the tSNE projection is increased after co-expression simulation, with no effect on cell-level identity.

6. This comment is about how automated and complete the package is. The authors mentioned that this is an "end-to-end" pipeline multiple times -- not quite sure what "end-to-end" means here, but the pipeline does involve manual steps seen from both the manuscript and the vignette. A main one is when merging the isoform clusters, the "visibly redundant clusters" (from vignette) were merged using a manually entered list. (It is indeed acknowledged on page 5, lines 8-9 that the clustering approach is "semi-automated".) Also, although in the manuscript and Figure 1 it is described that the acorde workflow includes the functional annotation and functional analysis parts (which are the key modules to test whether the whole pipeline leads to biologically meaningful results), the vignette stops at the step of returning DIU and coDIU genes, and it doesn't seem that the github package integrates the functions for functional analysis.

We sincerely thank the reviewer for taking the time to review our package and documentation -in addition to the manuscript- and for their constructive criticism.

We always intended to make our analytical decisions transparent and reproducible, and it is with that intention that we have included all details on any manual steps performed during our analyses in the package's vignette. Even though the final merge step can be easily automated by supplying adequate parameters to the `merge_clusters()` function, we decided to refine results manually to improve the detection of DIU and co-DIU genes, which is more robust with a smaller, less redundant number of clusters and, consequently, enhance our functional analysis. We believe that obtaining meaningful biological results sometimes requires manual inspection and, with that in mind, we documented some manual steps to improve the clustering results, given that full-automation often comes at the cost of less optimal results.

Nonetheless, taking into account the concerns raised by the reviewer and aiming to incorporate best practices into our analyses, we have slightly modified our clustering

analysis to make it substantially more automated. Therefore, with the exception of one manual merge and the re-assignment of isoforms from three noisy clusters, we now perform all required steps using the functions provided by *acorde*. Of note, to obtain results comparable to those generated via manual merge, we performed a second sequence of filter - expand - merge steps of isoforms in our clusters. Details are included in the Methods section as well as pages 9-10 of the results. We believe that these changes have made our results more reproducible and understandable and, since all steps are additionally documented in the vignette, users will get a better idea on how to combine the different functions in *acorde* for the optimal clustering of their data.

Regarding functional analysis, we have used some of the outputs of the last step in the *acorde* vignette, particularly DIU and coDIU gene lists, to perform a functional analysis in tappAS. tappAS' results have then been analyzed using R. In the light of the reviewer's comment, we have expanded the current vignette to include instructions for performing a functional analysis. In particular, we now explain how to generate tappAS inputs using the outputs from *acorde* and give some pointers on how to run the analyses in our manuscript. Explaining how to run these in tappAS, however, is beyond the scope of *acorde*'s documentation, as this information is available both on tappAS documentation and in its manuscript. In summary, even though functional analyses are not a part of the *acorde* R package, but a series of steps required for result interpretation, we appreciate the reviewer's criticism in this aspect and believe that their feedback will contribute to make the pipeline more useful for the community while simultaneously making our analyses more transparent and reproducible.

Minor points:

1. Page 5, line 10: the authors used "re-defined" which indicates there exists a previously defined version of the terms DIU and coDIU but there are no references here.

Usage of the word "re-defined" refers to the implementation of Differential Isoform Usage analysis in the previously-published tappAS application, in which we have based both the DIU and co-DIU analyses in the manuscript. For clarity, we have changed "re-defined" to "defined".

2. Page 5, line 19: by "cell type-specific co-expression of isoforms ...", do the authors mean "cell type-level co-expression of isoforms ..." or something else? "cell type-specific co-expression of isoforms" would mean isoforms that co-expression in certain cell types but not in other cell types to me, which is certainly interesting, but isn't seem to be performed in this work.

Thank you for bringing this to our attention. We have changed the phrase to "cell type-level co-expression of isoforms" in the newly submitted version of the manuscript.

3. Page 18, line 18: the authors use "oligodendrocyte-specific" and "neuron-specific" here (or oligodendrocytes vs neurons) and at other locations of the manuscript. It would be helpful for the readers if the authors can specify which of the 7 cell types defined on pages 6 and 7 each of these two categories (oligodendrocytes and neurons) correspond to.

We have edited the functional analysis section of our manuscript's results to ensure that readers can understand which cell types are specifically involved in this analysis.

Reviewer #2 (Expertise: splicing, scRNASeq):

In this manuscript, Arzalluz-Luque et al proposed a computational pipeline, acorde, to identify isoform co-usage network from single-cell transcriptomics data with a newly proposed percentile correlation method and to perform functional analysis of the clusters of genes with similar co-usage patterns. The splicing isoform analysis in single cells is an interesting topic; this paper has a focus on the functional analysis of co-used isoforms instead of the splicing's impact on cell heterogeneity.

The acorde R package could be a useful pipeline, but the novelty on the methodology in each step is not high. Instead, this manuscript is more a demonstration that the proposed analysis strategy of isoform co-usage can reveal biological functions, in a context of cell types, which can also be highly valuable. However, this manuscript is only based on one dataset, so it is hard to evaluate the generalizability of the proposed pipeline, particularly given there are multiple settings and thresholds to tune in each step of the pipeline. Therefore, it would be more insightful to apply the pipeline to more datasets and demonstrate the usefulness of these analyses, even without long-reads data to pre-filter the isoform annotations.

Following the reviewer's suggestion, we have included the analysis of a second dataset in the present version of the manuscript. The dataset consists in a subset of the data published in Tasic et al. 2018 (<https://www.nature.com/articles/s41586-018-0654-5>) in which we have selected GABA-ergic neuron cells and explored isoform co-expression patterns among the different subtypes detected in the original study. Results are included in Supplementary Figure 18 and pages 22-23 of the manuscript, whereas analysis decisions and the parameters used have been included in the last section of Methods.

The manuscript is well written and very detailed. Here, I have a few comments on the technical part:

1. The proposed percentile correlation is an appealing metric to measure the correlation between transcripts across cell types. From my understanding, it is an extension of median value of cells in a cell type. Instead of only using median value (a single percentile), this method uses 10 percentile values to represent the isoform expression in each cell type for calculating a Pearson Correlation Coefficient (PCC) of two isoforms across all cell types. Therefore, a more relevant comparison is to the median value of each cell type, so it can show how much the percentile correlation improves from median correlation, and it can imply their relation. From the Supp Fig. 3C, it seems 4-percentile is comparable to 10-percentile, so single-percentile (median) may be not too far.

We appreciate the reviewer's interest in our metric and, in order to further illustrate the improvement that percentile correlation provides in comparison to median (i.e. single-

percentile) correlation, we have added Supplementary Figure 2D, in which we depict the density distribution of Pearson correlation when cell type median expression is used as a proxy for isoform expression, and discussed the effect of single-percentile correlation in page 9, lines 13-16. We find that over-summarizing isoform expression creates a highly bimodal distribution in which most isoform-to-isoform correlations yield near-zero and near-one values. We believe this to be a consequence of the high zero-inflation in single-cell data, which translates into a high abundance of zero median values and, as a result, generates unstable correlations. Therefore, we recommend using more than one percentile value to better grasp cell type isoform expression, however sparse it may be.

2. Since the sample size ($n_{\text{cell_types}} * n_{\text{percentiles}}$) is small, the $PCC > 0.8$ may not necessarily mean a statistical significance. A permutation test would be useful to evaluate the significance, for example by shuffling the cell types of one of the two isoform, e.g., for a thousand times. I would suggest keep the percentiles of each cell type when shuffling them.

To ensure the validity of our metric's results, we have evaluated the significance of the percentile correlations obtained in the Tasic dataset. Taking advantage of the many tests that have been designed for this purpose, we applied the *cor.test()* function (R stats package) to percentile-summarized isoform expression in order to see which isoform pairs were significantly correlated, aiming to combine correlation value and AdjPValues as complementary criteria for clustering. However, our results show that there is a majority of low correlations that return significant AdjPValues, while all percentile correlations > 0.8 are also significant (see figure 2 above). We therefore maintain that percentile correlation value is a sufficiently reliable criteria to detect correlated isoform pairs, even more so when correlation is used as a distance metric for hierarchical clustering.

3. Related to point 2, in the simulated data (Supp. Fig. 5E), it only reports the proportion of highly correlated isoform pairs but didn't mention false positives. It could be possible that the percentile correlation is a method with high sensitivity, but also has high false positive rate.

Related to what was stated above, given the high abundance of significant correlations, we believe that using stringent thresholds for correlation values is an effective method to eliminate false positives. We would like to thank the reviewer for raising this point and helping us to critically assess our methodology and better demonstrate the reliability of our analyses. As a side note, we would like to stress that we have implemented a downstream ANOVA test to validate functionally-relevant co-DIU gene pairs, ensuring that another source of information other than correlation is provided for users to detect co-splicing patterns while reducing the number of false-positive co-DIU gene pairs.

4. In p.7, the number of detected DE isoforms are 10,100 out of 13,832, which means 73% transcripts are differentially expressed in at least one cell type comparing to all others. This proportion seems very high. It is important to evaluate how the initial selection of DE isoform impact on the downstream analysis. For example, one can add fold change as additional cutoff besides FDR as often used in DEseq or edgeR and use the overlapped DE isoform instead of the union between the two methods.

Thank you for this observation. Regarding the number of differentially expressed genes, although it might be high, we believe that it reflects the transcriptional diversity between cell types. We think that using the union, rather than the intersection, to select our initial set of transcripts, is a more suitable approach, as it allows us to detect more potential isoform expression changes. Given that we have introduced additional criteria to confidently identify isoforms with coordinated changes in subsequent steps, we believe this DE selection results in a reliable final isoform network.

However, as suggested by the reviewer, we have evaluated fold change as well as absolute and relative expression properties for all DE transcripts. To characterize these aspects of the Tasic dataset, we have represented the maximum fold change (Log2FC) between cell types (Figure 3) in the light of its relationship to maximum cell type absolute expression (i.e. maximum cell type mean). Regarding log2FC patterns, we saw no relationship to mean or median cell type expression, with DE transcripts combining patterns of low expression and high Log2FC with high expression and low Log2FC. A closer look at this pattern revealed that barely any transcripts (9 out of 10100) showed maximum Log2FC between cell types lower than 1, indicating $FC > 2$ for at least two cell types for all DE transcripts.

Figure 3: Relationship between Fold Change and cell type-level transcript expression. The x-axis represents the maximum Log2FC detected between the 7 broad cell types in the Tasic et al. 2016 neural dataset. The y-axis includes the maximum mean counts (Log2) among analyzed cell types. Each dot represents one Differentially Expressed transcript in the originally submitted manuscript (total: 10100). Colors represent the maximum median counts among cell types for each of the transcripts.

Importantly, we did detect a group of transcripts in which high FC was accompanied by very low mean expression and zero median counts in all cell types, i.e. those accumulating in the

lower right area of the plot in Figure 3. In line with the reviewer's concern, we considered these to be potentially noisy transcripts that had escaped zero-value filtering during QC. Specifically, we believed these to be transcripts in which moderate expression in a small proportion of the cells might be leading to significant differential expression. These changes, in spite of being significant, may not be sufficiently strong to consider them for Differential Isoform Usage at the gene level.

To account for this while also considering the behavior of all same-gene isoforms, we investigated the relationship of Log₂FC and mean cell type expression with the transcript's gene-relative expression (Figure 4). We detected a linear relationship between maximum gene-relative expression across cell types and transcript maximum mean expression. In addition, we saw that most transcripts accumulating a low proportion of the expression of their gene also showed large maximum Log₂FC and small cell type mean expression. We therefore decided to remove this small set of transcripts in order to improve the reliability of our alternative isoform co-expression analysis. In particular, we established a filter for minor isoforms, i.e. those whose expression constituted less than 10% of their gene's total expression (see details in Methods and in Results page 7). However, please consider that we have repeated our analyses with new Differential Expression criteria (see also point 3 in the response to reviewer #1). Therefore, even though the figures above represent analysis of the DE transcripts in the previously submitted version of this manuscript, the minor isoform filter has been applied to the set of DE isoforms used in our new analysis.

Figure 4: Relationship between gene-relative and cell type-level transcript expression. The x-axis represents the maximum gene-relative expression detected between the 7 broad cell types in the Tasic et al. 2016 neural dataset. Gene-relative expression of a transcript is understood as the proportion of an isoform's expression relative to the sum of expression of all the isoforms of said gene. The y-axis includes the maximum mean counts (Log₂) among analyzed cell types. Each dot represents one Differentially Expressed transcript in the originally submitted manuscript (total: 10100). Colors represent the maximum Log₂FC detected among cell types for each of the transcripts.

Finally, considering the insight obtained in these analyses, we decided to include a series of functions in *acorde* that implement filtering based on these three characteristics (cell type mean/median expression, gene-relative expression and fold-change) to allow users to perform this type of feature-level quality control.

5. The clustering of the isoform based on the correlation matrix is of high uncertainty because of both the noise in the correlation matrix and the selection of the number of clusters. This may be the reason why the authors introduce a semi-automated clustering, to adjust the clustering manually. Therefore, it is important to evaluate how the settings in this clustering step will impact on the downstream analysis. For example, based on the current setting (p.14), 80% of genes have differential isoform usage, which seems too high. So whether a smaller number of isoform clusters may give more stringent DIU.

We thank the reviewer for raising this concern. We agree that, indeed, the choice of parameters can impact the results of the clustering, however, the pipeline is designed to be flexible enough so that users can inspect their data and select those parameters that lead to optimal clustering. In this review, we have introduced several changes to minimize the uncertainty of clustering, including 1) checking potential biases in the selection of DE transcripts, such as fold-change, absolute expression and the relative importance of an isoform in its gene's expression (figures 3 and 4 in this response letter); 2) evaluating correlation p-values to ensure that no high, but non-significant correlations are considered for clustering (see Figure 2 in this letter) and 3) removing both the random downsampling of cells and the manual merge of clusters in our pipeline.

Regarding the semi-automated nature of the pipeline, we would like to stress that, as shown by our corrected results (pages 9-10 of the revised manuscript), manual curation of clusters can easily be replaced by additional refinement steps in the clustering pipeline, all of which are implemented as functions in *acorde*. However, even if non-automated steps are more costly, automation can sometimes lead to less accurate results, as can be seen when comparing manually-refined clusters in our previously submitted manuscript (figure 3B, previous version) and current clusters obtained with more automated steps (figure 2D, current version). Related to this, although it is possible to apply more lenient similarity criteria to group clusters, thus obtaining a smaller number of total clusters, this may make clusters noisier and lead to less reliable co-DIU relationships. We would like to point out that, to mitigate this, our package includes a way to test co-DIU pairwise relationships between genes in order to use statistical significance as an additional criteria to select genes for further functional analysis. Ultimately, however, these are all analytical decisions that depend on the aim of the analysis, and we believe that we have made our best to reflect that both in the manuscript and in the documentation of our package.

Finally, to clarify, the % of DIU genes was originally computed vs the total number of clustered isoforms. To give the correct %, these actually represent 23% of total genes that have at least one isoform in the expression matrix (post-QC). We have included the correct percentage in the manuscript (page 14, line 6).

6. Related to Fig. 5D where the number of coDIU has a linear relation to the number of pairs. It will be useful to report the significance on the n_{coDIU} compared to random by a permutation test, where two group of isoforms are randomly generated to a matched sizes of the two isoform clusters, and n_{coDIU} under the null can be calculated. By repeating this procedure many times, an empirical p value can be obtained. Both the permutation p values and the distribution of the n_{coDIU} under the null can be informative to present.

We would like to start by thanking the reviewer for proposing this analysis. Upon reading all reviewer's comments, we thought that it would be interesting to explore how many and which coDIU genes would be detected under a random (i.e. null distribution) scenario.

Following the reviewer's suggestion, we randomly distributed all clustered isoforms into a set of 15 clusters matching the sizes of those obtained via correlation-based clustering. We then computed coDIU genes, i.e. pairs of genes with matching isoform assignment. Of note, random assignment with no regard for isoform expression profiles precluded testing of coDIU significance between pairs of genes. Therefore, we compared coDIU pairs obtained after random clustering with the set of true coDIU pairs obtained, considering them prior to significance-based filtering.

First, we found that randomly assigning isoforms to clusters resulted in ~100K pairwise coDIU relationships between genes, while correlation-based clustering yielded slightly less than 80K coDIU pairs, reducing the number of pairs by more than 20%. Given that the number of genes is much larger than the number of clusters, finding such a high number of coDIU relationships is not a surprising result, even when grouping isoforms randomly. More importantly, however, we checked whether the same pairwise relationships were reproduced by the random clusters. In other words, we searched the ~80K true coDIU pairs and computed the % of relationships that were maintained after random clustering. Remarkably, we observed that only 9.6% of coDIU relationships were maintained among the randomly generated clusters. Of note, since no expression-related criteria was used to create isoform groups, it was unlikely that the same pairs were produced by random grouping.

Altogether, given that clustering is performed based on sound correlation and expression criteria (as opposed to random grouping) and considering that randomly generated coDIU pairs barely recapitulate the real results, we believe that this type of validation would not be adequate to measure coDIU significance. We have therefore refrained from including the analysis in the manuscript, however, we would like to remind the reviewer that our R package *acorde* includes a way of testing the significance of pairwise coDIU relationships (see Methods).

Reviewers' Comments:

Reviewer #1:

Remarks to the Author:

I would like to thank the authors for making the effort to address my comments. Most of my concerns were addressed. Following the comment regarding using a long-reads-based reference for mapping: I have one further concern. The authors have justified the choice of using the long-reads based reference genome, but this seems to have limited the application scope of acorde. The authors should make the following points clear so that users will easily understand whether acorde can be used for their needs:

How many long-reads based reference genomes acorde will incorporate? If it only incorporates the mouse brain data and the pipeline is made specifically for this cell type, it should be made clear somewhere obvious in the paper like in the abstract. Alternatively, the authors may consider providing other options to allow users to analyze data from other cell types and species with acorde: providing options of using standard reference genomes or providing curated long-reads based references for other cell types/species.

Reviewer #2:

Remarks to the Author:

I thank the authors for addressing to my comments. As an analysis pipeline, there are often multiple steps, like arcade here, so one common challenge is how the threshold choices in each step would affect the final results & biological discovery or interpretation. From this point of view, some of my comments, e.g., point 4, haven't been addressed directly regarding the impact on final outcomes, but their indirect clarification helps. Other than that, I don't have more technical comments, and my overall assessment remains the same as the initial version.

Reviewer #1 (Remarks to the Author):

I would like to thank the authors for making the effort to address my comments. Most of my concerns were addressed. Following the comment regarding using a long-reads-based reference for mapping: I have one further concern. The authors have justified the choice of using the long-reads based reference genome, but this seems to have limited the application scope of *acorde*. The authors should make the following points clear so that users will easily understand whether *acorde* can be used for their needs:

How many long-reads based reference genomes *acorde* will incorporate? If it only incorporates the mouse brain data and the pipeline is made specifically for this cell type, it should be made clear somewhere obvious in the paper like in the abstract. Alternatively, the authors may consider providing other options to allow users to analyze data from other cell types and species with *acorde*: providing options of using standard reference genomes or providing curated long-reads based references for other cell types/species.

We would like to thank the reviewer for taking the time to evaluate the revised version of our manuscript. Regarding remaining concerns related to the applicability of *acorde* to other reference transcriptomes, note that, although we have shown the benefits of using a long read-defined transcriptome, the method can be applied to any isoform-level quantification matrix. Therefore, users may use any standard reference transcriptome or, alternatively, choose to define their own reference using long read data. The only limitation -provided that they wish to perform downstream functional analyses- concerns the annotation of functional features for those isoforms.

In our case, we used the IsoAnnotLite tool to transfer functional labels from tappAS' pre-annotated mouse RefSeq 78 reference and from the mouse neural transcriptome used in the tappAS publication, which was possible thanks to the fact that we used a compatible reference for long-read processing (i.e. an updated version of the RefSeq transcriptome). As mentioned prior, details on how to perform the functional annotation step are included both in the Supplementary Note and in the R package vignette. While our lab is currently working on a tool for *de novo* functional annotation, which is expected to be released soon, users will need to rely on pre-annotated references and on isoAnnotLite for now. In the meantime, all available reference functional annotations, including the one used in the present manuscript, can be viewed and downloaded at: <https://app.tappas.org/resources/downloads/gffs/>

To make sure that users have all required information to use *acorde*, we have edited our package's README to include a «Getting started» section, in which we clarify these aspects of *acorde*'s applicability and point users to the right sites to obtain pre-computed GFF3 references.

Reviewer #2 (Remarks to the Author):

I thank the authors for addressing to my comments. As an analysis pipeline, there are often multiple steps, like *arcde* here, so one common challenge is how the threshold choices in each step would affect the final results & biological discovery or interpretation. From this point of view, some of my comments, e.g., point 4, haven't been addressed directly regarding the impact on final outcomes, but their indirect clarification helps. Other than that, I don't have more technical comments, and my overall assessment remains the same as the initial version.

Thank you for reevaluating our revised manuscript. It is true that choosing the right parameters is often difficult, as the reviewer points out. However, we trust that the extensive documentation that we have provided with the *acorde* package, in addition to the Methods section in the manuscript, will provide enough insight for future users to decide on the right parameters for their data.